# Insight and Recommendations for Fragile X-Premutation-Associated Conditions from the Fifth International Conference on *FMR1* Premutation

**DOI:** 10.3390/cells12182330

**Published:** 2023-09-21

**Authors:** Flora Tassone, Dragana Protic, Emily Graves Allen, Alison D. Archibald, Anna Baud, Ted W. Brown, Dejan B. Budimirovic, Jonathan Cohen, Brett Dufour, Rachel Eiges, Nicola Elvassore, Lidia V. Gabis, Samantha J. Grudzien, Deborah A. Hall, David Hessl, Abigail Hogan, Jessica Ezzell Hunter, Peng Jin, Poonnada Jiraanont, Jessica Klusek, R. Frank Kooy, Claudine M. Kraan, Cecilia Laterza, Andrea Lee, Karen Lipworth, Molly Losh, Danuta Loesch, Reymundo Lozano, Marsha R. Mailick, Apostolos Manolopoulos, Veronica Martinez-Cerdeno, Yingratana McLennan, Robert M. Miller, Federica Alice Maria Montanaro, Matthew W. Mosconi, Sarah Nelson Potter, Melissa Raspa, Susan M. Rivera, Katharine Shelly, Peter K. Todd, Katarzyna Tutak, Jun Yi Wang, Anne Wheeler, Tri Indah Winarni, Marwa Zafarullah, Randi J. Hagerman

**Affiliations:** 1Department of Biochemistry and Molecular Medicine, School of Medicine, University of California Davis, Sacramento, CA 95817, USA; mzafarullah@ucdavis.edu; 2MIND Institute, University of California Davis, Davis, CA 95817, USA; bdoufour@ucdavis.edu (B.D.); drhessl@ucdavis.edu (D.H.); vmartinezcerdeno@ucdavis.edu (V.M.-C.); 3Department of Pharmacology, Clinical Pharmacology and Toxicology, Faculty of Medicine, University of Belgrade, 11129 Belgrade, Serbia; dragana.protic@med.bg.ac.rs; 4Fragile X Clinic, Special Hospital for Cerebral Palsy and Developmental Neurology, 11040 Belgrade, Serbia; 5Department of Human Genetics, Emory University School of Medicine, Atlanta, GA 30322, USA; emily.g.allen@emory.edu (E.G.A.); peng.jin@emory.edu (P.J.); katharine.elizabeth.shelly@emory.edu (K.S.); 6Victorian Clinical Genetics Services, Royal Children’s Hospital, Melbourne, VIC 3052, Australia; alison.archibald@vcgs.org.au; 7Department of Paediatrics, Faculty of Medicine, Dentistry and Health Sciences, The University of Melbourne, Melbourne, VIC 3052, Australia; claudine.kraan@mcri.edu.au; 8Genomics in Society Group, Murdoch Children’s Research Institute, Royal Children’s Hospital, Melbourne, VIC 3052, Australia; 9Department of Gene Expression, Institute of Molecular Biology and Biotechnology, Adam Mickiewicz University, Uniwersytetu Poznańskiego 6, 61-614 Poznan, Poland; anna.baud@amu.edu.pl (A.B.); katarzyna.tutak@amu.edu.pl (K.T.); 10Central Clinical School, University of Sydney, Sydney, NSW 2006, Australia; wtbibr@aol.com; 11Fragile X Association of Australia, Brookvale, NSW 2100, Australia; karenlipworth@yahoo.com; 12NYS Institute for Basic Research in Developmental Disabilities, New York, NY 10314, USA; 13Department of Psychiatry, Fragile X Clinic, Kennedy Krieger Institute, Baltimore, MD 21205, USA; budimirovic@kennedykrieger.org; 14Department of Psychiatry & Behavioral Sciences-Child Psychiatry, School of Medicine, Johns Hopkins University, Baltimore, MD 21205, USA; 15Fragile X Alliance Clinic, Melbourne, VIC 3161, Australia; jcohen@travelclinic.com.au; 16Department of Pathology and Laboratory Medicine, Institute for Pediatric Regenerative Medicine, Shriners Hospitals for Children of Northern California, School of Medicine, University of California Davis, Sacramento, CA 95817, USA; yamclennan@ucdavis.edu; 17Stem Cell Research Laboratory, Medical Genetics Institute, Shaare Zedek Medical Center Affiliated with the Hebrew University School of Medicine, Jerusalem 91031, Israel; rachela@szmc.org.il; 18Veneto Institute of Molecular Medicine (VIMM), 35129 Padova, Italy; nicola.elvassore@unipd.it (N.E.); cecilia.laterza@unipd.it (C.L.); 19Department of Industrial Engineering, University of Padova, 35131 Padova, Italy; 20Keshet Autism Center Maccabi Wolfson, Holon 5822012, Israel; lidiagabis@gmail.com; 21Faculty of Medicine, Tel-Aviv University, Tel Aviv 6997801, Israel; 22Department of Neurology, University of Michigan, 4148 BSRB, 109 Zina Pitcher Place, Ann Arbor, MI 48109, USA; sgrudz@umich.edu (S.J.G.); petertod@med.umich.edu (P.K.T.); 23Neuroscience Graduate Program, University of Michigan, Ann Arbor, MI 48109, USA; 24Department of Computational Medicine and Bioinformatics, University of Michigan, Ann Arbor, MI 48109, USA; 25Department of Neurological Sciences, Rush University, Chicago, IL 60612, USA; deborah_a_hall@rush.edu; 26Department of Psychiatry and Behavioral Sciences, School of Medicine, University of California Davis, Sacramento, CA 95817, USA; 27Department of Communication Sciences and Disorders, Arnold School of Public Health, University of South Carolina, Columbia, SC 29208, USA; hoganbro@mailbox.sc.edu (A.H.); klusek@mailbox.sc.edu (J.K.); 28RTI International, Research Triangle Park, NC 27709, USA; jehunter@rti.org (J.E.H.); snpotter@rti.org (S.N.P.); mraspa@rti.org (M.R.); acwheeler@rti.org (A.W.); 29Faculty of Medicine, King Mongkut’s Institute of Technology Ladkrabang, Bangkok 10520, Thailand; poonnada.ji@kmitl.ac.th; 30Department of Medical Genetics, University of Antwerp, 2000 Antwerp, Belgium; frank.kooy@uantwerpen.be; 31Diagnosis and Development, Murdoch Children’s Research Institute, Melbourne, VIC 3052, Australia; 32Fragile X New Zealand, Nelson 7040, New Zealand; andrea@fragilex.org.nz; 33Roxelyn and Richard Pepper Department of Communication Sciences and Disorders, Northwestern University, Evanston, IL 60201, USA; m-losh@northwestern.edu; 34School of Psychology and Public Health, La Trobe University, Melbourne, VIC 3086, Australia; d.loesch@latrobe.edu.au; 35Departments of Genetics and Genomic Sciences and Pediatrics, Icahn School of Medicine at Mount Sinai, New York, NY 10029, USA; reymundo.lozano@mssm.edu; 36Waisman Center, University of Wisconsin-Madison, Madison, WI 53705, USA; marsha.mailick@wisc.edu; 37Intramural Research Program, Laboratory of Clinical Investigation, National Institute on Aging, Baltimore, MD 21224, USA; apostolos.manolopoulos@nih.gov; 38National Fragile X Foundation, Washington, DC 20005, USA; robby@fragilex.org; 39Child and Adolescent Neuropsychiatry Unit, Department of Neuroscience, Bambino Gesù Children’s Hospital, IRCCS, 00165 Rome, Italy; alicemontanaropsicologa@gmail.com; 40Department of Education, Psychology, Communication, University of Bari Aldo Moro, 70121 Bari, Italy; 41Schiefelbusch Institute for Life Span Studies, University of Kansas, Lawrence, KS 66045, USA; mosconi@ku.edu; 42Clinical Child Psychology Program, University of Kansas, Lawrence, KS 66045, USA; 43Kansas Center for Autism Research and Training (K-CART), University of Kansas, Lawrence, KS 66045, USA; 44Department of Psychology, University of Maryland, College Park, MD 20742, USA; smrivera@umd.edu; 45Ann Arbor Veterans Administration Healthcare, Ann Arbor, MI 48105, USA; 46Center for Mind and Brain, University of California Davis, Davis, CA 95618, USA; jyiwang@ucdavis.edu; 47Center for Biomedical Research (CEBIOR), Faculty of Medicine, Universitas Diponegoro, Semarang 502754, Central Java, Indonesia; triwinarni@lecturer.undip.ac.id; 48Department of Pediatrics, School of Medicine, University of California Davis, Sacramento, CA 95817, USA

**Keywords:** *FMR1* premutation, FXPAC, FXTAS, FXAND, FXPOI, *FMR1* molecular and clinical

## Abstract

The premutation of the fragile X messenger ribonucleoprotein 1 (*FMR1*) gene is characterized by an expansion of the CGG trinucleotide repeats (55 to 200 CGGs) in the 5’ untranslated region and increased levels of *FMR1* mRNA. Molecular mechanisms leading to fragile X-premutation-associated conditions (FXPAC) include cotranscriptional R-loop formations, *FMR1* mRNA toxicity through both RNA gelation into nuclear foci and sequestration of various CGG-repeat-binding proteins, and the repeat-associated non-AUG (RAN)-initiated translation of potentially toxic proteins. Such molecular mechanisms contribute to subsequent consequences, including mitochondrial dysfunction and neuronal death. Clinically, premutation carriers may exhibit a wide range of symptoms and phenotypes. Any of the problems associated with the premutation can appropriately be called FXPAC. Fragile X-associated tremor/ataxia syndrome (FXTAS), fragile X-associated primary ovarian insufficiency (FXPOI), and fragile X-associated neuropsychiatric disorders (FXAND) can fall under FXPAC. Understanding the molecular and clinical aspects of the premutation of the *FMR1* gene is crucial for the accurate diagnosis, genetic counseling, and appropriate management of affected individuals and families. This paper summarizes all the known problems associated with the premutation and documents the presentations and discussions that occurred at the International Premutation Conference, which took place in New Zealand in 2023.

## 1. Introduction

The discovery and sequencing of the fragile X messenger ribonucleoprotein 1 (*FMR1*) gene [1] have led to new molecular testing to facilitate the diagnosis of those with fragile X syndrome (FXS) with >200 CGG repeats, and the methylation of the promoter and the repeats located within the 5′UTR of the gene. Carriers of the premutation (PM) were found to have 55 to 200 CGG repeats, did not have methylation, could pass on the full mutation to their offspring, and were presumed to be unaffected because *FMR1* protein (FMRP) levels were usually normal. Males with the PM were called “non-penetrant and transmitting males” because they were thought to be unaffected and passed on the PM to their daughters without the repeat expanding to the FM range. The PM term reflected the lack of clinical involvement, and this concept was soon to crumble. In this introduction, we outline the historical progression of PM research and present the current state of the science in an effort to provide context for the emerging findings presented and for the dynamic discussion held at the 2023 International Fragile X Premutation Conference.

Even before the discovery of the *FMR1* gene, four women, who had sons with FXS, attending a National Fragile X Foundation (NFXF) conference luncheon in 1987, surprised the others at the table, including scientists, as they all spoke about early menopause in their 30 s. In a subsequent survey, 104 female carriers were divided into those that had an IQ less than 85 vs. greater than or equal to 85. Thirteen percent of carriers with an IQ of 85 or above were found to have an early menopause versus 0% of those with an IQ < 85 and 5% of the normal controls. Although this finding did not quite reach statistical significance, it suggested that carriers with an average or greater IQ (who later turned out to have the PM) had an increased prevalence of early menopause [2]. Subsequent studies have confirmed the presence of fragile X-associated primary ovarian insufficiency (FXPOI) in PM carriers, which is associated with a bell-shaped relationship with the CGG repeat number; those with repeats between 85 and 100 have the highest risk and earliest onset of FXPOI [3,4,5]. Drs. Flora Tassone and Paul Hagerman discovered elevated levels of the *FMR1* mRNA in PM carriers compared to controls, the opposite of what was expected. The blood of carriers had between two and eight times the normal values of the *FMR1* mRNA, with a positive association with the CGG repeat number in the PM range [6]. The same year, at the NFXF meeting in Los Angeles in 2000, the Hagerman team presented case summaries from five aging male carriers with a history of tremor, ataxia, and atrophy using magnetic resonance imaging (MRI), and these cases were published in 2001 [7]. The researchers thought that this was a rare finding; however, when the family audience, which included over 100 carriers, were asked if they knew of relatives with similar problems, about 50% raised their hands, leading to a multitude of studies documenting the phenotype of what was later known as the fragile X-associated tremor/ataxia syndrome (FXTAS). The name of FXTAS and the original diagnostic criteria were established with the description of over 40 cases, as reported in Jacquemont et al. [8]. The awareness of FXTAS was dramatically improved with another paper published in JAMA documenting the prevalence of tremor and ataxia in carriers utilizing all the families identified in California at that time [9]. They found that the incidence of FXTAS increased with age in male carriers; 17% in their 50 s had tremor and balance problems, but this number gradually increased with age, such that 75% had tremor and ataxia in their 80 s. The researchers also found that females had fewer motor symptoms than males [9].

FXTAS is now well recognized as a neurodegenerative disorder with tremor, ataxia, neuropathy, and parkinsonian features, as well as cognitive changes beginning with memory problems and executive function deficits [10,11,12,13,14,15,16]. Additionally, MRI findings of white-matter disease, usually in the middle cerebellar peduncles (MCP sign) and periventricular areas, in addition to the splenium of the corpus callosum [17], have been documented. Neuropathological studies have demonstrated the presence of intranuclear inclusions in both neurons and astrocytes [18], and more recently, the enhanced activation and frequent death of astrocytes [14], iron overload [19], frequent microbleeds [20], parkinsonian features, including loss of dopamine cells, and occasional Lewy-body inclusions [21]. Eventually, 50% of males with FXTAS develop dementia [22], but this is far less common in females with FXTAS [23].

The pathophysiology of FXTAS involves multiple mechanisms, including RNA toxicity [24,25,26], clogging of the proteasome [27], RAN translation [28,29], and mitochondrial dysfunction [30,31,32] (for more details, see Section 2, ‘The molecular basis of FXPAC’). Recent papers have shown that males progress more rapidly in motor symptoms than females, presumably because of the protective effects in addition to the presence of the normal second X chromosome [33]. Therefore, the phenotype of FXTAS appears to be somewhat different in females, but emotional problems, such as anxiety and even pain symptoms, are more common in females than in males, and these problems progress faster in females [23,33,34].

The expanded phenotype beyond FXPOI and FXTAS in female carriers dates to the study by Coffey and colleagues [35] who studied 128 non-FXTAS adult female carriers and 18 women with FXTAS compared to age-matched controls [35]. The authors found multiple medical conditions, including neuropathy, hypertension, autoimmune thyroid disease, chronic muscle pain, intermittent tremor, and fibromyalgia, that were significantly increased in carriers compared to controls, and many of these issues were seen in carriers without FXTAS. These findings have led to further studies of problems that occur in carriers before the onset of FXTAS and of disorders that can occur even in childhood in a subgroup of carriers. Although most carriers have normal intellectual abilities and are without neuropsychological issues, studies have shown that a subgroup of carriers have psychiatric problems in childhood, including anxiety [36], attention deficit hyperactivity disorder (ADHD) [37,38], social deficits [39], and even autism spectrum disorder (ASD) [37,40,41,42]. For carriers who experience seizures, there is a higher incidence of ASD or intellectual disabilities (ID) compared to carriers without seizures [43], and 20% of carriers with ID and ASD have a second genetic hit, as detected with whole-exome sequencing (WES) or microarray studies [44].

Chen et al. [45] have demonstrated that PM neurons die more easily in culture, leading to the concept that they may be more vulnerable to environmental toxins, as seen in the cellular studies of Song et al. [46], who studied the effects of several toxins. In the clinical realm, we see that exposure to isoflurane in general anesthesia can lead to the onset of FXTAS after surgery in elderly carriers [47]. In addition, toxic substances, such as illicit drugs, opioids, and excessive alcohol consumption, can lead to the more rapid progression of FXTAS [48,49,50,51,52]. Furthermore, research suggests that lifestyle changes to avoid toxins, environmental exposures, adverse experiences, and illnesses, such as diabetes, vitamin deficiencies, or hypothyroidism, may be helpful to slow down the progression of FXTAS [53].

It is likely that the pathophysiological changes in carriers, including mitochondrial dysfunction [30,31,54] and calcium dysregulation [55], can occur well before FXTAS and lead to GABA deficits [56], chronic pain [34], chronic fatigue [57,58], increased stress [59], mental health problems, and sensitivity to environmental stimuli [60]. In addition, several medical problems occur more frequently in carriers of the PM compared to the general population, such as autoimmune diseases [61], hypertension [62], insomnia [57], migraines [63], and connective tissue problems [64], which can rarely present as sudden coronary artery dissection (SCAD) [65] and cardiac arrhythmias [66]. Recognition of these findings will likely lead to further research and treatment endeavors [53]. Medication trials in FXTAS are described under the FXTAS treatment section of this review paper.

Mental health impact has been documented particularly in female carriers compared to controls over the last two decades, including anxiety, depression, obsessive–compulsive behavior, ADHD inattentive type, and the broad autism phenotype [67,68,69] (reviewed in [60]). Roberts et al. [70] have reported that psychiatric symptoms can become more common with age in adulthood. Women have expressed that their physicians do not take their concerns seriously and basically blame these psychological problems on the stress of raising a child with FXS, even though these problems can be seen in carriers without children or without children with FXS [71,72]. Although many scientists doubted that psychological/psychiatric problems could be related to the PM, the work of Marsha Mailick and colleagues has validated some of these findings [73]. They studied the Marshfield cohort of over 20,000 patients and conducted *FMR1* genotyping on the sample, but the patients and clinicians were naive to the results of the DNA testing. This research found elevated rates of agoraphobia, social anxiety or social phobia, and panic disorder, but not higher rates of major depression episodes in the medical records database in the male and female carriers compared to male and female noncarriers. This study demonstrated a higher prevalence of anxiety conditions in an unbiased group of people with the PM from the general population, as smaller studies have previously shown. A strong argument for the association between the PM status and psychological/psychiatric problems in female carriers was provided by the finding of highly significant (nonlinear) negative correlations between the size of CGG repeats and a great majority of SCL-90-R subscale scores and all the global indices [74].

The psychological difficulties can be severe and can occur in up to 50% of adult carriers. The name, fragile X-associated neuropsychiatric disorders (FXAND), was coined as an umbrella term to encompass the problems that are increased in carriers compared to controls, and are listed in the DSM5 [60]. Johnson et al. [75] have objected to the term FXAND because there are milder mental health impacts that do not meet the criteria for a disorder, so they proposed the term fragile X-PM-associated conditions (FXPAC), avoiding the use of the term “disorder”. Thus, the various physical and mental conditions mentioned above, and any of the problems associated with the PM, can appropriately be called FXPAC so that the more specific and detrimental PM issues, such as FXAND, FXPOI, and FXTAS, can fall under this category.

The goal of this paper is to document the presentations and discussions that occurred at the International Premutation Conference covering all the known problems associated with the PM. This conference took place in a wonderful location in the North Island of New Zealand, where we learned about the amazing new research presented in this paper and in the dedicated volume of *Cells.*

## 2. The Molecular Basis of FXPAC

The PM alleles are characterized by increased levels of *FMR1* mRNA, which correlate with the length of the repeat tract, in both male and female carriers of a PM allele [6,76,77]. Although the elevated mRNA levels result from an increase in transcriptional gene activity [78], a CGG-repeat-length-dependent decrease in the expression of the *FMR1* protein, FMRP, likely results from the impaired scanning of ribosomal preinitiation complexes through CGG-repeat tracts [6,76,79,80]. The increased expression of the *FMR1* mRNA (up to six-to-eight-fold of that seen in normal alleles) leads to transcriptionally activated cellular stress pathways, RNA-mediated toxicity triggering CGG-binding-protein sequestration, and repeat-associated non-AUG-initiated (RAN) translation, which are the current basic and central molecular mechanisms proposed to explain the pathogenesis of FXTAS.

### 2.1. Molecular Basis of the FMR1 Locus

The PM alleles in females are unstable and prone to expansion on intergenerational transmission, with expansion into alleles harboring greater than 200 CGG repeats, leading to FXS. Generally, one or two AGG interruptions are observed within the repeat tract of normal and intermediate *FMR1* alleles (6–44 CGG and 45–54 CGG repeats, respectively), while one or none are observed in PM alleles, and they are known to influence the stability of the repeats during parental transmission. Specifically, the presence of AGG interruptions decrease the intergenerational instability of the CGG repeats, thus decreasing the risk of expansion to a full-mutation allele [81,82]. In addition to AGG interruptions, other factors that increase the risk of expansion to full-mutation alleles during maternal transmission include the maternal CGG repeat number and age [81,82]. Interestingly, no association was found to correlate with either the transcriptional or translational activity of the gene [78,80,83,84].

As observed in other trinucleotide disorders, a bidirectional transcription at the *FMR1* locus has been demonstrated and specific alternative splicings of the antisense *FMR1* (*ASFMR1*) gene have been identified [85]. The *ASFMR1* gene is expressed in all tissues, with high expression observed in the brain, spans approximately 59 kb of genomic DNA, and contains 13 exons and 45 *ASFMR1* isoforms that are identified, 19 of which are expressed only in the PM [86]. Some of these isoforms are, as for the *FMR1* gene, highly expressed in the PM as compared to the controls [87]. Although the *ASFMR1* has been suggested to play a critical role in the pathogenesis of FXTAS [88,89], further studies are warranted to shed light on the contribution of the *ASFMR1* in the clinical phenotypes of FXTAS.

Recently, it has been demonstrated that alleles in the PM range can be somatically unstable in both male and female carriers of a PM allele [86,90]. As observed with intergenerational instability, it was demonstrated that the extent of somatic instability directly correlates with the number of CGG repeats and, inversely, with the number of AGG interruptions. Increased levels of somatic expansion are observed over time in blood (PBMCs) derived from female carriers of a PM allele [86], and are mainly due to unmethylated *FMR1* alleles and, therefore, limited to the active X chromosome. Recent evidence suggests that DNA repair factors *FAN1* and *MSH3* are both also modifiers of the expansion risk in females with specific genotypes associated with increased somatic instability [86] (these genes have also been implicated in other repeat expansion disorders (Genetic Modifiers of Huntington’s Disease Consortium)), suggesting a common expansion mechanism. Genetic factors that affect the somatic expansion risk may contribute to the variable penetrance for FXPAC that is seen. The extent of somatic instability in female PM carriers has shown a significant correlation with a diagnosis of ADHD [91], and may also affect the risk of various PM conditions in both males and females.

Allelic instability, observed in individuals with *FMR1* mutations, leads to both intra- and intertissue mosaicism (PBMCs, fibroblasts, and brain tissues), and may account for some of the variability observed in the clinical phenotype of individual carriers of the PM [90]. During the International Premutation Conference, new data were presented about allelic instability within the *FMR1* gene, confirming its occurrence between and within different tissues derived from the same individuals. Unstable alleles were exhibited among the majority of both female and male PM carriers. In addition, diverse allele profiles were displayed between PBMCs and fibroblasts from the same individuals among PM males, in accordance with previous studies [90,92,93,94,95]. Allelic instability affirms the complexity of *FMR1* mutations and may relate to diverse phenotypes, including cognitive abilities and behavioral features observed in both FXS and PM disorders [96], specifically in female carriers of a PM allele with ADHD and depression [91]. The activation ratio (AR) is a clinically relevant parameter for females with both full-mutation [97] and PM conditions [86], as it reflects the fraction of normal alleles present on the active X chromosome [98]. The X-inactivation process is widely recognized as a factor that can influence the symptoms and severity of many diseases [99]. In FXS, although the size of the CGG repeat in the promoter region of the *FMR1* gene is a significant factor, it is not sufficient to entirely determine the functionality of the gene. Hence, factors such as the AR and methylation status of the gene in females carrying an *FMR1* mutation may also contribute to the regulation of FMRP levels. Therefore, to accurately interpret phenotypic characteristics in individuals with both FXS and FXPAC, it is necessary to assess methylation-status analyses [100,101,102,103].

The extent of phenotypic variation based on the AR is demonstrated by the observation that approximately 30% to 50% of females carrying a full mutation and exhibiting normal intelligence have the mutation primarily on their inactive X chromosome [103]. Moreover, studies have indicated that female PM carriers with a higher AR exhibit a significantly lower FXTAS incidence [35,104,105]. On the other hand, individuals with a normal allele that is predominantly methylated, and therefore inactive, may be at a higher risk of developing FXTAS. Additionally, several studies have suggested that lower AR values could be linked to cognitive and behavioral challenges in female PM carriers [97,106,107,108,109,110], potentially affecting the risk, severity, and age of onset of FXPAC. Despite numerous studies investigating the role and impact of the AR in PM carriers, there are discrepancies among their findings that may be partially attributable to technical variability, as previously reported [86,111,112,113,114], or to differences in the methods employed to calculate the AR (as discussed in Protic et al., this special issue [115]). At the International Premutation Conference, novel data were presented demonstrating a noteworthy correlation between clinical measures and the AR. As anticipated, the study revealed that higher ARs were linked to reduced *FMR1* transcript levels for any given repeat length, and associated with enhanced performance, verbal, and full-scale IQ scores, as well as lower levels of depression, and a smaller number of medical conditions. Based on this evidence, it is advisable to evaluate the methylation status, including the AR in females with both PM and full-mutation alleles of the *FMR1* gene, to better understand their clinical phenotypes.

### 2.2. Molecular Mechanisms Leading to FXTAS Pathology—RNA Toxicity and RAN Translation at CGG Repeats: Mechanistic Insights and Their Contribution to Disease Pathology

There are currently three nonexclusive models for how CGG repeats elicit pathogenesis in FXTAS (Figure 1).

In one, CGG-repeat RNAs elicit a gain-of-function toxicity through both RNA gelation into nuclear foci and sequestration of various rCGG-repeat-binding proteins [25,26]. Mass-spectrometric and immunohistochemical analyses have identified over 20 proteins in the frontal cortex inclusions of FXTAS patients, including RNA-binding proteins (RBPs), HNRNP A2/B1 (heterogeneous nuclear ribonucleoprotein A1), and MBNL1 (muscleblind-like protein 1), as well as some neurofilament proteins, such as lamin A/C and α-internexin. These proteins are involved in various neurological disorders [119]. Pur α and HNRNP A2/B1 bind directly to rCGG repeats in inclusions, and their overexpression in a *Drosophila* model expressing PM CGG repeat expansions suppressed neurodegeneration phenotypes [25,26]. Sequestration of other proteins, such as CUGBP1 (CUGBP Elav-like family member 1), SAM68 (Src-associated substrate during mitosis of 68-kDa), Rm62 (ATP-dependent RNA helicase p62), and DGCR8 (DiGeorge syndrome critical region 8), leads to altered mRNA splicing and transport, as well as dysregulated microRNAs, supporting a toxic RNA gain-of-function mechanism mediated by the expanded CGG repeats in *FMR1* [24,26,120,121,122].

HNRNPA2/B1 is present in intranuclear inclusions of FXTAS patients and it binds directly to rCGG repeats. Its overexpression, along with its two homologs in *Drosophila*, suppresses the neurodegenerative eye phenotype caused by the rCGG repeat [26]. HNRNP A2/B1 also mediates the indirect interaction between CGG repeats and CUGBP1 involved in myotonic dystrophy type 1 (DM1). Overexpression of CUGBP1 suppresses the FXTAS phenotype in *Drosophila*. Pur α, another protein found in intranuclear inclusions of FXTAS patients, plays a crucial role in DNA replication, neuronal mRNA transport, and translation. Pur α knockout mice show developmental delays and altered expression, and the distribution of axonal and dendritic proteins [123,124]. Overexpression of Pur α in a *Drosophila* model suppresses rCGG-mediated neurodegeneration in a dose-dependent manner. Sequestration of SAM68 in particular causes pre-mRNA alternative splicing misregulation in CGG-transfected cells and FXTAS patients, thus contributing to FXTAS pathogenesis via a splicing alteration mechanism [120]. TDP-43 (TAR DNA-binding protein 43), an ALS-associated RBP, has reduced association with ribosomes in the cerebellar Purkinje neurons of mice expressing 90 CGG repeats [125]. In the same study, the authors went on to find that, in the *Drosophila* model of FXTAS, wild-type TDP-43 expression leads to suppression of neurodegeneration, while the knockdown of the endogenous TDP-43 fly ortholog, TBPH, enhanced the eye phenotype.

Another study also independently reported that TDP-43 suppressed CGG-repeat-induced toxicity in a *Drosophila* model of FXTAS [126]. Interestingly, this suppression was shown to depend on HNRNP A2/B1, such that the deletion of the C-terminal domain of TDP-43, and thereby the prevention of interactions with HNRNP A2/B1, led to the abrogation of the TDP-43-dependent rescue of CGG repeat toxicity [126]. Finally, DGCR8, a protein binding to PM rCGG repeats, causes partial sequestration of DGCR8 and its partner, DROSHA (drosha, ribonuclease 3), within PM RNA aggregates. DGCR8 and DROSHA play a critical role in microRNA biogenesis. Sellier and colleagues found that the sequestration of DGCR8 and DROSHA precludes them from their normal functions, leading to reduced processing of pri-miRNAs in cells expressing expanded CGG repeats. Consequently, levels of mature miRNAs are also reduced in the brains of FXTAS patients [24].

Alternatively, the CGG repeats in 5′ UTR of *FMR1* mRNA may be translated into toxic proteins through a process known as RAN translation. Initially described at CAG repeats in spinocerebellar ataxia type 8 (SCA8) and DM1 [127], noncanonical translation of short tandem repeats into proteins may occur in the absence of an AUG initiation codon when repeat-containing RNAs form stable secondary structures. RAN translation has been observed on repeats associated with ten disorders: SCA8, DM1, DM2, HD, FXTAS, C9orf72 amyotrophic lateral sclerosis and frontotemporal dementia (C9 ALS/FTD), FXPOI, SCA31, and Fuchs endothelial corneal dystrophy (FECD) (reviewed in [118,128]). In many of these diseases, RAN translation occurs in different reading frames on both sense and antisense transcripts, and the RAN products are detected in patient tissues.

In FXTAS, it is thought that CGG repeats form secondary structures that lead to the impairment of ribosomal scanning, reduced start codon fidelity, and, in consequence, aberrant translation initiation at near-cognate or noncognate codons located upstream or within the repeats [129]. Depending on the reading frame, different toxic proteins containing long mono-amino-acid tracts are produced: polyglycine (FMRpolyG), polyalanine (FMRpolyA), and polyarginine (FMRpolyR) [28,129]. Additionally, there is evidence that RAN translation also can occur on the CCG antisense transcript [130] to produce additional homopolymeric proteins. Translation through the repeat may also trigger frameshifting to produce chimeric RAN proteins [131]. The translation of FMRpolyG is the most efficient, and this protein is detected in FXTAS patient brains by both immunohistochemistry and mass spectrometry, colocalizing with p62 and ubiquitin-positive inclusions [11,18,28,119,130,132,133,134,135]. However, quantitation of this and other RAN-translation-generated proteins remains challenging due to their low abundance, solubility, multiple initiation sites, and early translation termination—all of which hamper its detection by antibodies targeting either the N- or C-terminus [21,134,135]. FMRpolyG was found to interact with the nuclear lamina protein LAP2β, leading to the impairment of the nuclear lamina architecture [132]. Additionally, FMRpolyG is capable of cell-to-cell propagation via exosomes in cell-culture studies and glia-to-neuronal propagation in mouse-model systems. Similar prion-like propagation is thought to play a central role in the pathogenic spread of alpha synuclein in Parkinson’s disease and of Tau in Alzheimer’s disease (PMID: 30917002). However, the role of this phenomenon in FXTAS pathogenesis remains unclear [134,136].

Whether RAN products generated from CGG repeats are drivers of toxicity or if there is instead a synergy between CGG-repeat RNA and RAN proteins remains unknown. Studies in overexpression systems in cells, flies, and mice suggest that near-cognate codons 5′ to the repeat that support the RAN translation of FMRpolyG are requisite to elicit maximal toxicity [28,132,137,138]. However, FMRpolyG inclusions can persist even as phenotypes resolve when the repeat is transcriptionally silenced [139]. Moreover, FMRpolyG production absent the repeat RNA is less toxic in neurons than is a RAN-competent CGG repeat [131].

The exact mechanism by which RAN translation occurs remains enigmatic, and may vary in different repeats (and even different reading frames of the same repeat). However, several recent studies reported modifiers of RAN translation that provide some clues. Unwinding the structured RNA is crucial for RAN translation, as it is shown that several RNA helicases, such as DDX3X (ATP-dependent RNA helicase DDX3X), DHX36 (ATP-dependent DNA/RNA helicase DHX36), eIF4A/B (eukaryotic initiation factor 4A-I/B), and H, are directly involved in the regulation of this process enabling proper ribosomal scanning [129,137,140]. In addition, the presence of RAN proteins, together with structured RNAs with CGG repeats, leads to the activation of the integrated stress response (ISR) and the phosphorylation of eIF2α, which, in a feed-forward-loop mechanism, shut down the global translation but selectively enhance RAN translation [141]. Proteins which interact with CGG-repeat RNAs may also influence the RAN translation, as SRSF1 (serine/arginine-rich splicing factor 1) mediates the nuclear retention of CGG-repeat RNAs to prevent these transcripts from becoming a template for RAN translation [118].

### 2.3. Therapeutic Perspectives to FXTAS from a RAN-Translation Perspective

There are currently no FDA (Food and Drug Administration Agency)-approved drugs to slow FXTAS progression or delay its onset. An emphasis point that was raised during the International Premutation Conference was that there is a critical need for the discovery of reliable, robust biomarkers to accurately understand predisease onset states and readouts for clinical progression. Some promising work suggests that metabolomic and/or proteomics biomarkers may serve this purpose [142,143,144]. Indeed, a small open-label pilot study in patients with validation studies in patient fibroblasts indicated that the mitochondrial activator sulforaphane showed some correction of these biomarkers that could serve as a precursor for a larger study [143].

Antisense oligonucleotides (ASOs) hold promise for FXTAS treatment by blocking RAN translation in neurons without degrading the *FMR1* mRNA, enhancing FMRP expression, and improving neuron survival [138]. In rodent models, ASOs reduce FMRpolyG biosynthesis, correct disease-related traits, and normalize transcriptomic effects [145]. A recent study suggests that the ubiquitin–proteasome system may be an interesting therapeutic target based on the presence of PSMB5 (proteasome subunit beta type-5) polymorphisms as disease-onset modifiers in patients and the suppression of disease-relevant phenotypes in *Drosophila* with a genetic knockdown of this proteasomal subunit [146]. This factor also changes how CGG RAN translation happens in cell tests. For example, it affects how a molecule called CMBL4c can attach to CGG-repeat RNA structures and lower FMRpolyG levels [147]. The ISR targeted by protein kinase R (PKR) rescue in a mouse model of FXTAS [143], along with multiple CGG-repeat-associated RBPs, offer potential treatments targeting RAN-translation modifiers, although clinical application awaits further research due to the early stage of *Drosophila*, cell-based, and mouse experiments.

### 2.4. Genetic Modifiers in Fragile X-Associated Tremor Ataxia Syndrome (FXTAS)

The underlying neurobiological mechanisms of FXTAS are complex and not fully understood. As mentioned above, several mechanisms that have been proposed to explain the pathogenesis of FXTAS, including RNA toxicity, RAN translation producing the accumulation of the FMR PolyG polypeptide, and damage response, are linked to white-matter-tract connectivity in the brain, called white-matter hyperintensities, and strongly associated to the clinical impairment observed in FXTAS [6,28,148]. However, not all individuals who carry a PM allele will develop PM conditions, including FXTAS, in their older adulthood, which indicates the incomplete penetrance pattern of the disease. Therefore, nowadays, some studies have been dedicated to a plausible mechanism and exploring predisposing factors, including genetic modifiers, that may contribute to the occurrence of FXPAC. Investigations of genetic modifiers of clinical manifestation of diseases have also become a new research interest in FXTAS. They sought to provide an answer to the wide diversity and severity of clinical major criteria (intention tremor and gait ataxia) and minor criteria (cognitive impairment) [33,146,149]. Various genetic variants may contribute to cognitive impairment, including the APOe4 allelic variant, which represents the strongest risk factor of late-onset Alzheimer’s disease (AD), the most common type of dementia, in all ethnic groups [150]. The prevalence of the APOe4 allele is 13.7% in the general population; having one copy of the APOe4 allele increases the risk by around 3 times compared to individuals without the APOe4 allele, while having two copies boosts the risk of AD by 8–12 times [151].

APOE is an important cholesterol and lipid transporter that plays a critical role in a variety of signaling pathways in the development, maintenance, and repair mechanisms of the central nervous system (CNS) [152]. The APOe4 allele triggers β-amyloid (Aβ) accumulation/amyloidosis in oligodendrocytes and their myelin that leads to the slowing of brain electrical signaling, which is associated with cognitive impairment [153]. Postmortem examination of FXTAS brain tissue showed the presence of cortical amyloid plaques and neurofibrillary tangles, combined with presence of intranuclear inclusions in those with FXTAS and AD, which is additional evidence of the involvement of other genes that may modify the FXTAS phenotype [154]. Among the *FMR1* PM carriers, the APOe4 allele frequency is higher (31.8%) in patients with FXTAS compared to the general population and increases the risk by more than 12 times to develop the disease [155]. During the International Premutation Conference, data on 180 PM males, aged over 50 years, were presented which showed that the APOe4/APOe2 and APOe4/APOe3 genotypes were more frequent in PM males with FXTAS compared to those without FXTAS (2% vs. 0% and 10.6% vs. 2.4%, respectively).

Recently, to identify the genetic modifiers of FXTAS, a large number of PM carriers were recruited for whole-genome sequencing (WGS), which was further combined with *Drosophila* genetic screening. It was demonstrated that using FXTAS *Drosophila* as a genetic screening tool can be powerful in the validation of candidate genes from WGS. Eighteen genes were identified as potential genetic modifiers of FXTAS. One of such candidate genes is the proteasome subunit beta-5 (PSMB5), which genetically modulates CGG-associated neurotoxicity in *Drosophila* as a strong suppressor of CGG-associated neurodegeneration. PM individuals who carry the variant PSMB5rs11543947-A, which is associated with decreased expression of PSMB5 mRNA, may be protected against FXTAS. In addition, there is a strong suppression of CGG-associated neurodegeneration through diminishing RAN translation in the *Drosophila* knockdown of PSMB5 [146]. The metabolomic approach to determine a genetic modifier in an FXTAS mouse model found metabolic changes and demonstrated that Schlank (ceramide synthase), Sk2 (sphingosine kinase), and Ras (IMP dehydrogenase), which encode enzymes in the sphingolipid and purine metabolism, respectively, were significantly related with FXTAS-CGG-associated neurodegeneration pathogenesis [149].

Finally, more studies are needed to identify possible genetic modifiers associated with FXTAS development and progression for better management of the disease and for the development of therapeutic strategies.

### 2.5. The Use of Human Pluripotent-Stem-Cell-Based Neurodevelopmental Models for FXTAS

Human models of FXPAC are essential tools for studying disease-specific mechanisms, such as RNA toxicity, RAN translation, and CGG somatic instability. However, generating improved model systems for all these pathologies requires patients’ disease-relevant cell cultures. In the case of FXTAS, this is especially challenging because postmortem brain samples are rarely available, limited to a small amount of biological material, and represent only the final stage of the disease.

Overcoming these limitations can be achieved by utilizing mutant human pluripotent stem cells (hPSCs) in conjunction with in vitro differentiation towards affected tissues (neurons). This approach provides a powerful tool for both fundamental and applied research, offering an excellent opportunity to investigate the disease’s pathogenic mechanisms and to identify potential targets for therapeutic intervention.

There are two types of pluripotent stem cell lines that can be utilized for FXTAS disease modeling: human embryonic stem cell (hESC) lines derived from genetically affected embryos that can be obtained by preimplantation genetic diagnosis (PGD) procedures [156], and patient-derived induced pluripotent stem cells (iPSCs), established by reprogramming somatic cells obtained from patients (e.g., blood, skin fibroblasts) [157]. Both PGD-derived hESCs and patient-derived iPSCs carry the disease-causing PM and can reproduce disease cellular phenotypes in vitro, and allow following dynamic processes that are misregulated during development and aging in patients. However, most of the literature is based on the use of human iPSC-derived in vitro models, and very little has been done using hESCs [158,159], so we will focus only on iPSC-based in vitro models.

The first in vitro model of FXTAS using pluripotent stem cells (PSCs) showed that differentiated neurons from iPSCs recapitulate the cellular phenotypes of FXTAS, including reduced synaptic puncta density, neurite length, and increased calcium transients [157]. FXTAS iPSCs were also used to discover a toxic mechanism linked to FMRPolyG proteins via RAN translation [132]. Additionally, human neurons derived from patient iPSCs were used to validate a therapeutic approach that selectively blocked CGG RAN initiation sites using noncleaving ASOs. ASO blockade improved endogenous FMRP expression, suppressed repeat toxicity, and prolonged survival in human neurons, showing the therapeutic potential of modulating RAN translation in FXTAS [138].

Nevertheless, despite recent progress, the currently available human iPSC-based models for FXTAS are insufficient in reproducing the full complexity of the disease. This is because these models are based on monolayer cell cultures, which restrict the analysis to less-mature and single-cell types. To gain a comprehensive understanding of the interactions between various cell populations in the brain, and to examine the contribution of each pathogenic mechanism associated with FXTAS during early brain development, a higher level of complexity than monolayer cell cultures, such as brain organoids, would be necessary.

Brain organoids are three-dimensional mini-organs derived from PSCs that mimic the cellular composition and architecture of specific brain regions [160]. As such, they are expected to provide a powerful tool for identifying critical molecular events in the development of FXTAS, much before the clinical signs appear in patients. Moreover, brain organoids could extend our knowledge on other aspects of the disease, such as CGG somatic instability and the generation of mosaicisms for expansion size and/or methylation, in a multicellular setting that more closely resembles the developing human brain.

### 2.6. Shared Molecular Mechanism with Other Repeat Expansion Disorders

FXTAS is a repeat expansion disorder that displays clinical symptoms similar to those observed in other diseases caused by repeat expansions. Parkinsonism, a varied array of cognitive impairments that can progress to dementia, and amyotrophic lateral sclerosis (ALS)-like phenotypes, including frontotemporal dementia and progressive supranuclear palsy, have all been reported in FXTAS [161]. Tremor and ataxia, which are also hallmark symptoms of other repeat expansion disorders, such as spinocerebellar ataxias, are commonly observed in FXTAS.

The genetic basis of the FXTAS repeat expansion is similar to other repeat expansions observed in several diseases, including C9orf72 ALS/frontotemporal dementia (GGGGCC-repeat), myotonic dystrophy type 1 (CTG-repeat), NOTCH2NLC (CGG-repeat), Huntington’s disease (CAG-repeat), and spinocerebellar ataxias (SCA-CAG-repeat). Regional aggregation of cytosolic, nuclear, or extracellular proteins is a common feature observed in these diseases and disrupts neuronal function [162]. Intranuclear eosinophilic ubiquitin-positive inclusions in neurons and astrocytes are characteristic of FXTAS pathology and have been observed in other trinucleotide disorders [163]. TDP-43 in ALS/frontotemporal dementia and poly (amino acid)/polypeptides in FXTAS, Huntington’s disease, and spinocerebellar ataxias are examples of the types of aggregates that result from the expansion of trinucleotide repeats [164].

The most common genetic cause of ALS/frontotemporal dementia is an expanded GGGGCC-repeat in the C9orf72 gene. Similar to FXTAS, RAN translation and the accumulation of toxic peptides in neurons and astrocytes (TDP-43) are the main pathological mechanisms in C9orf72 ALS/frontotemporal dementia [165]. The accumulation of toxic polypeptides resulting from expanded trinucleotide repeats is also observed in Huntington’s disease (CAG-repeat) and spinocerebellar ataxias (CAG-repeat) [166].

The NOTCH2NLC pathogenic CGG-expansions, located in the 5′ UTR (66-517) and having GGA or AGC interruptions, are particularly similar to those observed in FXTAS. They cause a late-onset disorder with a clinical variability that includes muscle weakness, dementia, parkinsonism, tremor, and ataxia. The molecular mechanisms of OTCH2NLC lead to neuronal intranuclear eosinophilic inclusions, and the antisense isoform has been hypothesized to be a pathological mechanism [167].

Anticipation, somatic instability, and clinical severity associated with the number of repeats has been described in many repeat expansion disorders, including, HD, DM1, FXTAS, ALS, and others [168].

Aside from these, FXTAS resembles DM1 in many respects. Firstly, because the primary mechanism for both pathologies is RNA toxicity [25,26,132,169,170,171,172]. Secondly, and as mentioned above, both affected loci exhibit RAN translation potential, leading to the production of toxic polyglycine, polyalanine and polyarginine containing proteins by CGG expansion in the PM range in *FMR1* [28,132,173], and polyalanine- and polyserine-containing proteins by CTG expansions in DM1-affected cells [127,174]. To add further complexity, both disorders exhibit a decrease in protein levels, albeit through distinct mechanisms [6,76,175]. Lastly, both expansions in *FMR1* and DM1 display maternal anticipation/expansion, giving rise to distinct phenotypes (namely, FXS in *FMR1* and congenital myotonic dystrophy type 1 in DMPK) and to DNA hypermethylation. Altogether, the clinical presentation of individuals carrying the *FMR1* PM is highly heterogeneous and shares similarities with the phenotypic heterogeneity observed in DM1 and other nucleotide repeat disorders. This variability likely results from the involvement of the multiple mechanisms that, together with modifier genes and environmental factors, contribute to disease pathology to varying degrees.

### 2.7. Mitochondrial Dysfunction in PM Carriers

Recently, studies on cultured cell lines, animal models, and human subjects have implicated mitochondrial dysregulation in the pathogenesis and progression of FXTAS. Using magnetic resonance imaging (MRI), Rizzo et al. (2006) [176] first described lactate accumulation in the lateral ventricles, as well as decreased ATP levels in the calf muscles of a patient with FXTAS. Subsequent studies on cultured fibroblasts from PM carriers and mouse models have confirmed impaired ATP production and the pathogenic role of expanded CGG repeats on mitochondrial functions [177,178]. Finally, clinical studies on living patients with FXTAS and postmortem brain tissues with the disease have showed altered Krebs-cycle intermediates, neurotransmitters, and neurodegeneration markers, as well as reduced mitochondrial DNA copy numbers in specific brain regions, such as the cerebellar vermis, parietal cortex, and hippocampus [32,179]. Finally, unlike the earlier results from human brain tissue, studies in Epstein–Barr-virus (EBV)-transformed blood lymphoblasts showed that mitochondrial respiratory activity was significantly elevated in FXTAS compared with controls. Specifically, altered complex I activity and ATP synthesis, accompanied by an altered mitochondrial mass and membrane potential, were observed, and were significantly associated with the white-matter-hyperintensity (WMH) scores in the supratentorial regions [180]. In addition, an elevation of AMP combined with the reduction of TORC in both FXTAS and non-FXTAS categories of PM carriers was reported [181]. In the later study, correlations between measures of mitochondrial and nonmitochondrial respiratory activity, AMPK, and TORC1 cellular protein kinases, and the scores representing motor, cognitive, and neuropsychiatric impairments, were found with the CGG repeat size, and a hyperactivity of cellular bioenergetic components was significantly associated with motor-impairment measures, including tremor–ataxia, parkinsonism, and neuropsychiatric changes, predominantly in the FXTAS subgroup [182]. Moreover, an elevation of AMPK activity and a decrease in TORC levels were significantly related to the size of the CGG expansion. All the above studies have suggested that the bioenergetic changes in blood lymphoblasts are biomarkers of the clinical status of *FMR1* carriers. Furthermore, a decreased level of TORC1—the mechanistic target of the rapamycin complex—suggested a possible future approach to therapy in FXTAS.

Several molecular mechanisms have been proposed as mediators of abnormal mitochondrial function in FXTAS. RNA toxicity was the first model described, according to which the expanded CGG repeats in *FMR1* mRNA binds and titrates specific RNA-binding proteins, resulting in loss of their normal functions [26]. Among these proteins, the pre-mRNA splicing factor TRA2A has gained significant attention, since it is also present in the pathognomonic ubiquitin inclusions of FXTAS [183]. Additionally, miRNAs are increasingly recognized as major determinants of normal mitochondrial function. One of their biogenesis regulators, the DROSHA/DGCR8 enzymatic complex, was found sequestered within the expanded CGG RNA foci, leading ultimately to the loss of its normal function [24,184,185]. Moreover, altered zinc and iron metabolism, a pivotal neuromodulator, and an essential element in maintaining mitochondrial physiology, respectively, may be additional contributing factors in FXTAS pathogenesis. Fibroblasts from PM carriers have been shown to express abnormal zinc transporter levels, thereby leading to altered zinc homeostasis [30], whereas increased iron levels were also observed in neurons and oligodendrocytes of the putamen of carriers of a PM [19]. Finally, among the functions of FMRP, the product of the *FMR1* gene, is the binding to superoxide dismutase 1 (SOD) mRNA and the regulation of its levels. Consequently, lower expression of FMRP may result in decreased levels of SOD1, thereby leading to increased reactive oxygen species (ROS) levels and impaired oxidative phosphorylation [186].

More recently, emerging evidence has implicated the role of abnormal electron-transport-chain enzyme complexes in FXTAS pathogenesis. Gohel et al. had first observed defective complex activity in human cell lines and a transgenic mouse model [187]. Additionally, a recent study, presented at the International Premutation Conference, utilizing brain-derived extracellular vesicles, a novel and powerful platform for biomarker development for brain diseases, from plasma and from postmortem brain tissues from patients with FXTAS, found a decreased quantity and activity of complex IV and V, thus further validating this pathogenic process [143].

### 2.8. Omics Studies (Metabolomics and Proteomics) in PM Carriers

The development of targeted therapeutics for rare age-dependent neurodegenerative disorders encounters numerous challenges, encompassing the absence of biomarkers for early diagnosis and disease progression, intricate underlying molecular mechanisms, heterogeneous phenotypes, limited historical data, and the difficulties posed by conducting clinical trials with small patient populations, which restrict enrollment. In this context, contemporary omics studies, including metabolomics and proteomics, have emerged as promising tools for investigating global changes within a given sample, employing extensive data mining and bioinformatic analysis [188]. Recent advancements in metabolomic- and proteomic-profiling technologies and processing have enabled the efficient and precise analysis of several hundred metabolites/proteins, facilitating the identification of biomarkers associated with disease development and progression [189].

Giulivi et al. (2016) conducted a comprehensive analysis of the plasma metabolic profile in human PM carriers with FXTAS, comparing them to healthy noncarrier controls. Their findings identified a panel of four core serum metabolites (phenethylamine, oleamide, aconitate, and isocitrate) that exhibited high sensitivity and specificity in diagnosing PM carriers with and without FXTAS. Notably, the presence of oleamide/isocitrate was identified as a specific biomarker for FXTAS. Moreover, based on these plasma metabolic profiles, the researchers reported evidence of mitochondrial dysfunction, neurodegeneration markers, and proinflammatory damage in FXTAS PM carriers [32]. In a separate investigation, Song et al. (2016) reported increased mitochondrial oxidative stress in primary fibroblasts obtained from PM carriers compared to age- and sex-matched controls [46]. Napoli et al. (2016) examined peripheral blood mononuclear cells (PBMCs) derived from controls and carriers of a PM allele, with and without FXTAS, to investigate the presence of the Warburg effect. Their study revealed alterations in glycolysis and oxidative phosphorylation, indicating the involvement of the Warburg effect in FXTAS [190]. Using a PM murine model, Kong et al. (2019) investigated metabolic changes associated with FXTAS in the cerebellum. Their findings demonstrated significant alterations in sphingolipid and purine metabolism in the cerebellum of the mice. Furthermore, they identified genetic modifiers (Cers5, Sphk1, and Impdh1) of CGG toxicity in *Drosophila* [149]. In a 12-week open-label intervention study involving six males with FXTAS, Napoli et al. (2019) evaluated the effect of allopregnanolone on lymphocytic bioenergetics and plasma pharmacometabolomics. They observed the significant impact of allopregnanolone treatment on oxidative stress, the GABA metabolism, and certain mitochondria-related outcomes. These findings suggested the potential therapeutic use of allopregnanolone for improving cognitive function and the GABA metabolism in patients with FXTAS [191]. A more recent study by Zafarullah et al. (2020) aimed to identify metabolic biomarkers for early diagnosis and disease progression in FXTAS. Through characterization of individuals who developed FXTAS symptoms over time, alterations in the lipid metabolism, particularly in mitochondrial-bioenergetics-related pathways, were identified as significant contributors to FXTAS [88]. Subsequently, Zafarullah et al. (2021) established a significant correlation between the identified metabolic biomarkers and the area of the pons in individuals who developed FXTAS over time. They also demonstrated a notable association between these biomarkers and disease progression, highlighting their role within the context of the dysregulated lipid and sphingolipid metabolism [142].

In addition, the effort to identify the metabolic changes associated with FXPOI is ongoing, and preliminary data of a nontargeted metabolomic profiling of FXPOI patient plasma by LC/MS were presented during the International Premutation Conference. Initial differential abundance analyses revealed the altered abundance of compounds in the omega-6 fatty-acid (n-6 FA) metabolism and arachidonic acid formation between females with a FXPOI diagnosis and female carriers of a PM without POI across both cohorts. Pathways downstream of FA and the arachidonate metabolism were also identified, including prostaglandin synthesis and the formation of proinflammatory metabolites from the AA. Further investigation of metabolic changes associated with FXPOI is likely to provide critical information about the mechanism of dysfunction in PM ovaries.

In recent years, Ma et al. (2019) conducted an LC-MS/MS-based proteomics analysis of intranuclear inclusions isolated from the postmortem brain tissue of individuals with FXTAS. Their findings revealed the presence of over 200 proteins within the inclusions, with significant abundance of SUMO2 and p62/sequestosome-1 (p62/SQSTM1). These results support a model where inclusion formation is a consequence of increased protein loads and heightened oxidative stress [134]. Subsequently, Holm et al. (2020) characterized the proteomic profile of the FXTAS cortex compared to that of healthy controls (HCs). They observed a notable decrease in the abundance of proteins, such as tenascin-C (TNC), cluster of differentiation 38 (CD38), and phosphoserine aminotransferase 1 (PSAT1), in the FXTAS samples. Additionally, the authors confirmed a significantly elevated abundance of novel neurodegeneration-related proteins and small ubiquitin-like modifier 1/2 (SUMO1/2) in the FXTAS cortex compared to HCs [27]. Furthermore, Abbasi et al. (2022) reported changes in the level of multiple proteins, including amyloid-like protein 2, contactin-1, afamin, cell-adhesion molecule 4, NPC intracellular cholesterol transporter 2, and cathepsin, by comparing the cerebrospinal fluid (CSF) proteome of FXTAS patients with HCs. Alterations in acute-phase-response signaling, liver X receptor/retinoid X receptor (LXR/RXR) activation, and farnesoid X receptor (FXR)/RXR activation pathways were also observed [192]. In an ongoing study, the Tassone lab performed blood proteome profiling of PM-allele carriers who developed FXTAS over time and compared it to HC samples. Through this analysis, they identified potential proteomic biomarkers for early diagnosis and reported altered protein pathways between the groups, suggesting their involvement in the pathogenesis of the disorder [144]. However, due to the limitations of a small sample size, further studies with larger cohorts are necessary to validate the initial findings and elucidate the role of the identified markers and pathways.

### 2.9. CGG Short Tandem Repeat (STR) Expansions

It has been outlined that the molecular cause of FXTAS is the presence of a PM ranged (55–200 units) expansion of the CGG short-tandem-repeat (STR) locus located within the 5′-UTR of the *FMR1* gene [7]. In recent years, several other neurodegenerative disorders have been associated with a PM ranged CGG STR expansion as their genetic cause [193,194,195,196,197]. These diseases include neuronal intranuclear inclusion disease (NIID), oculopharyngodistal myopathy (OPDM), and oculopharyngeal myopathy with leukoencephalopathy (OPML). These PM expansion loci are localized within the following genes and ncRNA: LRP12 (OPDM type 1), GIPC1 (OPDM type 2), NOTCH2NLC (OPDM type 3/NIID), RILPL1 (OPDM type 4), and LOC642361 (OPML). All of these disorders share a striking level of clinical similarity with FXTAS, suggesting a shared or similar molecular mechanism of pathology leading to a neurodegenerative phenotype. In search of potential additional disease loci, Annear and colleagues (2021) performed a bioinformatic in silico analysis of the reference genome and identified approximately 6000 additional CGG STR loci. When large population datasets were analyzed (*n* > 12,000), 99% of these novel loci were demonstrated as displaying at least some degree of polymorphism across the human population, and approximately 15% of all CGG loci were observed to expand up to or beyond the 55-unit PM breakpoint [198]. How many of these loci may be involved in neurodegenerative disease remains an enigma. While the repeat length is unlikely the only factor affecting the pathogenic potential of a given repeat, it is no doubt a core component. Moreover, half of these CGG STRs displayed characteristics similar to the known disease-linked repeats [198]. This included high rates of polymorphism and a genetic localization within the 5′ UTR and gene promoter regions, a typical characteristic of disease-linked CGG STRs. However, there may be further factors at play, such as cis elements flanking the repeat and the reading frame of the repeat in reference to the localized gene [132,199]. In each case, it cannot be excluded that additional expansions of CGG STRs may play a role in progressive neurodegeneration disorders with FXTAS and FXPOI-like phenotypes. While additional expansions are not detected in routine diagnostics using current short-read-based detection methods, the future introduction of long-read sequencing may expose potential additional loci in the clinic.

Fragile X-premutation-associated-condition involvement across the lifespan is presented in Figure 2.

## 3. Clinical Involvement in Children Who Have a PM

Children and adolescents with a PM may present with clinical symptoms. As demonstrated at the conference, a key theme dominating this space is the increased nuance and understanding of the phenotype in children with a PM and how to manage it clinically.

Interest in the question of if, and how, a child with a PM is clinically impacted spans over a decade. Suggestions of increased risk of ASD, developmental characteristics, and speech and language disorders in children with a PM were some of the earliest observations [37,42]. It is not clear how common these are, though large-scale prevalence studies that have screened ASD and developmental-delay cohorts for the enrichment of children with PMs suggest that penetrance at the more severe end is uncommon [200,201,202,203]. Findings presented at the International Premutation Conference by Hunter and colleagues also demonstrated the likely rarity of children with this phenotype. In this presentation, the authors reported no difference in the proportion of children with a PM who fell in the clinically significant range on the parent-report standardized measures of behavior, emotional, and social outcomes [143]. The cohort described at the conference is one of the largest that this field has observed (88 PM males and 57 PM females) to investigate above-threshold neurodevelopmental outcomes in pre- and school-age children with PMs (age ~6 years). A strength of this study was that it recruited through prenatal diagnosis to minimize ascertainment bias. However, reliance on parent-report measures is a limitation, and more granular and comprehensive assessment of the early development of PM children is needed.

Interestingly, a new evidence-base is growing around the more nuanced clinical impacts of the PMs in childhood. Studies suggest that children with a PM may indeed have increased risk for sensory challenges [204], generalized anxiety, specific and social phobias, obsessive–compulsive disorder [36], and ADHD [205]. Clinical opinion is that learning difficulties that may impact school performance (esp. arithmetic difficulties), and subthreshold ASD traits are also elevated in children who have a PM. These outcomes largely map onto what is being observed in adult studies, providing additional evidence and adding validity to trends observed in the studies of children [73,206,207,208,209].

The findings presented by Hogan and colleagues at the International Premutation Conference have extended our understanding of the social-anxiety phenotype [143]. The presented data were from a small PM cohort (8 PM males and 11 PM females) ascertained through families with known family histories of FXS. Using highly targeted measures of social inhibition, which is a developmental precursor of social anxiety [210,211], and pragmatic language (i.e., social use of language), the authors showed that PM females aged ~4–7 years exhibited greater social inhibition than their age-matched peers. Pragmatic language abilities, however, were comparable between the two groups. Given that pragmatic language differences have been observed in adults with a PM [69,212,213], it remains unknown when in development these differences begin to emerge.

Taking previous literature and new directions from the International Premutation Conference, we suspect that most children with a PM have largely typical development and function. That said, our understanding of learning difficulties, subclinical symptoms, and neuropsychiatric presentations (which are harder to notice clinically, especially in early childhood) is emerging. Thus, we stress that, in the case of an identified child with a PM, we do still recommend that clinicians be cognizant about potential learning, behavioral, and psychiatric difficulties, even if the symptoms are below the threshold for clinical diagnosis. It was also noted in the conference discussion that, in children with a PM who have more severely affected siblings with FXS, these more subtle features are often overshadowed, as parents may be less aware of the ongoing challenges experienced by the child with the PM. However, with good clinical judgment and appropriate individualized assessment, treatment, and management options, long-term trajectories into adulthood may be improved or even optimized. Management options may include a developmental approach, cognitive-behavioral therapy (CBT), medications (specifically SSRIs), occupational and speech therapies, and/or behavioral strategies [214,215,216]. Current guidelines recommend both CBT and medications (specifically SSRIs) as first-line options for anxiety disorders. Other treatment options that could be explored are OT, speech–language therapy, behavioral strategies, and educational accommodations (such as extra time on exams or modified assignments).

Important emerging spaces to watch are described below:Increasing efforts to prepare support organizations, genetic counselors, and healthcare practitioners to be able to respond to and treat children who have a PM and who are symptomatic;Detailed characterization of the pediatric phenotype—both at clinically actionable and subthreshold levels;Efforts to study outcomes at a population scale through newborn screening that may provide an evidence-base around developmental trajectories and risks;Clarified testing indications and, potentially, modified diagnostic testing workflows to ensure that symptomatic children with PMs do not miss out on comprehensive genetic testing with microarrays and potentially other methodologies (WES or WGS).

In conclusion, based on the emerging literature and conference presentations, the growing consensus is that difficulties in sensorimotor and visuospatial processing, social inhibition, social anxiety/phobia, ADHD, and learning disabilities may manifest developmentally in some people with a PM. These children need to be offered appropriate individualized assessment, treatment, and management options to optimize outcomes. New knowledge about the characteristics of the phenotype is likely to impact testing indications within current genetic testing pathways, and the field has great hope that newborn screening studies can clarify questions about penetrance and developmental timing.

## 4. FXPAC and Relationships with Genetic Markers

### 4.1. FXTAS: Neurological/Cognitive Phenotypes

The original core motor features of FXTAS included cerebellar gait ataxia and intention tremor in *FMR1* PM men over the age of 60 [7]. Parkinsonism was also described, in addition to neuropathy, dysautonomia, and cognitive changes in the form of executive dysfunction progressing into dementia at the final stage of this disorder. The cerebellar gait ataxia of FXTAS typically appears after the onset of tremor and is progressive, resulting in falls and injury over time [217]. FXTAS patients have greater postural sway, with loss of balance control on posturography [218]. Eye-movement abnormalities associated with some other cerebellar disorders are rare. Although the findings of abnormal optokinetic nystagmus, slowed vertical saccades, and vertical gaze palsy, as well as square-wave jerks, were reported in isolated cases [219], a larger study with blinded neuro-ophthalmologist ratings did not show differences in ocular pursuit or saccadic dysmetria visible on neurological examination [220]. However, the eye-movement saccade-latency deficits, previously reported by [221], were replicated by [222] and in a study of women with FXTAS presented by Mosconi and colleagues at the International Premutation Conference.

It was not until the work of Grigsby and colleagues that a clearer picture of the cognitive phenotype associated with FXTAS was recognized through standardized neuropsychological assessments and specialized tests that measure the frontal/executive control of movement [223,224,225]. These studies revealed that, while verbal intelligence and domains of perceptual reasoning not involving motor coordination are relatively spared, measures assessing general mental status, the regulation of manual motor movements, verbal fluency, processing speed, temporal sequencing, working memory, inhibition, short-term memory, and cognitive flexibility tended to show significant deficits. These deficits were first characterized as a ‘dysexecutive’ syndrome [224]. In addition to motor and cognitive impairments, the high rate of psychiatric changes, such as anxiety and depression, were reported in both males and females affected with FXTAS [60,68]. The clinical features of FXTAS have been associated with the white-matter degeneration, largely involving the middle cerebellar peduncles, and visualized on MR images as the ‘MCP’ sign, which became one of the essential diagnostic criteria of this syndrome in males [8]. A detailed description of the MRI findings in FXTAS is provided below.

At the International Premutation Conference, a parallel between the constellation of the motor and cognitive dysfunction and psychiatric problems observed in FXTAS to the ‘cerebro-cerebellar cognitive affective syndrome, CCAS’, first described by Schmahmann et al. (1998) [226], was brought to the participants’ attention. In that syndrome, cerebellar damage, which was previously identified with motor dysfunction presenting as gait ataxia, dysmetria, tremor, and disordered eye movements, was linked with cognitive decline and psychiatric features. This constellation of changes can be explained by the close connection of the cerebellum with the cerebral cortex via the cerebro-cerebellar-cortical/limbic loops. However, the normally observed co-occurrence of (predominantly) cerebellar white-matter degeneration with cognitive, psychiatric, and motor changes does not necessarily imply a causative link. Instead, correlations between these domains are more informative, in that significant relationships indicate that these domains are likely to stem from the same pathogenic mechanism.

The relevant data providing evidence for such a relationship in males affected with FXTAS were presented by Loesch at this conference [143]. The study employed a battery of cognitive assessments, two standard motor rating scales, and two self-reported measures of psychiatric symptoms in a sample of 23 adult males > 50 years old affected with FXTAS. When controlling for age and/or educational level, where appropriate, there were highly significant correlations between the motor rating score for the ICARS gait domain and the scores representing global cognitive decline (ACE-III), processing speed (SDMT), immediate memory (Digit Span), and depression and anxiety scores derived from both SCL90 and DASS-21 instruments [227]. Significant relationships of most scores for three phenotypic domains with the size of the CGG repeat within the PM range suggested a close tracking with genetic liability. Remarkably, a similar pattern has been observed in a sample of 57 apparently asymptomatic adult female PM carriers [228,229].

Despite the regular occurrence of definite (syndromic) FXTAS in nearly half of the male, and about 14–16% of the female, PM carriers, there is a great (and still unexplained) diversity of clinical neurological manifestations both within and beyond this syndrome. Four different subphenotypes have been distinguished within FXTAS according to the type of tremor: (i) Intention tremor–cerebellar ataxia phenotype; (ii) Essential-tremor phenotype; (iii) Orthostatic-tremor phenotype; (iv) Rest tremor–parkinsonism phenotype [230], which is fully supported by the observations of Loesch, who discussed this issue at the International Premutation Conference. More specific information concerning the frequency of three of these tremor patterns was obtained earlier by applying clinical and electrophysiological methods. Essential-tremor-like tremors occurred in 35% of patients, intention/cerebellar tremors in 29% of patients, and resting/parkinsonian tremors in 12% of patients; 24% of patients showed no detectable tremors [231]. A parkinsonian phenotype, as observed in 64% of FXTAS patients, manifested as predominantly hypomimia and rigidity, with only a small proportion having a rest tremor [231,232]. This relatively large contribution of parkinsonism to the FXTAS phenotype is consistent with the findings from the [123I]-CIT SPECT (single-proton emission computed tomography) imaging, which showed a loss of presynaptic dopaminergic terminals with reduced putaminal uptake in a portion of FXTAS patients [233,234]. Generally, it was observed that some carriers had initially presented with tremor alone for more than a decade prior to developing other symptoms of FXTAS, and these carriers showed a more favorable disease course [232]. Additionally, carriers presenting with tremor alone have a lower rate of cognitive impairment, at 38%, compared to those with both ataxia and tremor at onset, at 68%. A similar phenomenon is seen in Parkinson’s disease (PD), where tremor-predominant PD is associated with fewer cognitive deficits than mixed or akinetic-rigid PD presentations [235].

Apart from the major risk factor of age, CGG repeat sizes higher than 70 within the PM range were shown to be associated with a greater risk of developing features of FXTAS [236], and lower repeat sizes within this range were shown to be correlated with the later onset of tremor and ataxia [78]. The distributions of the CGG repeat expansion size in the male sample of non-FXTAS versus FXTAS subjects show the peaks corresponding to these two respective carrier categories, implying that the middle range of repeat sizes (80–110) coincides with the highest risk of developing FXTAS, and the lower range to the non-FXTAS group [227]. In the same study, highly significant relationships have been reported (and demonstrated at the International Premutation Conference) between the tremor ataxia (ICARS), parkinsonism (UPDRS), and varieties-of-tremor (CRST) scale scores, as well as the overwhelming majority of cognitive and psychiatric dysfunction scores and the CGG repeat expansion size, in a sample of 28 FXTAS males. These data showed that the CGG repeat expansion size is predictive of the severity of the phenotype, as well as of the age of onset and the presence/absence of signs, rather than just the severity of the motor signs, as reported in earlier studies [11,219].

Age of death from FXTAS has been shown to correlate inversely with the repeat size [11,237]. However, despite the association with the age of death, the CGG repeat size did not correlate with the duration of the disease [237].

A large number of studies so far have concerned male carriers with FXTAS. As shown by the existing reports, this syndrome is much less frequent—and has a different profile and progression—in female compared with male carriers [23,34,238]. The 2021 study (Loesch et al.) comparing quantitative measures representing motor, cognitive, psychiatric and MRI changes in male and female carriers with FXTAS, showed a much lesser degree of cerebellar ataxia combined with an absence of the MCP sign—with more severe tremor and neuropsychiatric problems—in females. These results, which were also presented by Loesch at the International Premutation Conference, suggest the existence of unknown genetic modifiers, which may affect the clinical/neurological phenotype of females, in addition to the preventative and predictive effect of the second (normal) allele on the X chromosome. Indeed, the first evidence for genetic modifiers has been presented in the series of pioneering presentations at this conference [238].

Another presentation at the International Premutation Conference concerning female carriers was given by Berglund et al., who reported on the clinical features of a cohort of patients from Sweden. Interestingly, in their cohort, the women (*n* = 21) had an earlier onset of FXTAS (44–60 years) compared to the FXTAS men (*n* = 12, 49–64 years), despite having lower CGG repeats (and presumed X-inactivation) [143]. Penetrance of the disease was similar to previously reported studies [8], and Swedish women were more likely to have a ‘probable’ diagnosis compared with Swedish men, who were more likely to be diagnosed with ‘definite’ FXTAS. These findings in the Swedish population are consistent with other FXTAS cohorts, as are the racial and ethnic demographics.

### 4.2. FXTAS Spectrum: Nonsyndromic Neurological, Cognitive, and Psychiatric Involvements

The diversity of clinical involvement in PM carriers extends beyond a syndromic form of FXTAS. This issue has been raised at the International Premutation Conference: several examples of PM-associated neural involvement not meeting the FXTAS diagnostic criteria have been presented, and their implications in understanding the underlying pathological mechanisms have been discussed. In a major review of this aspect by Loesch, examples of mild neurological manifestations were reported, such as isolated ET-like intention tremors in male and female carriers from an Australian sample. Only a small proportion of these mild monosymptomatic forms converted to diagnosable FXTAS over an average of 8 years. Another notable example of the wide clinical spectrum of neural involvement in PM carriers was given in [33], where detailed neurological testing and scoring revealed the presence, and further progression, of subclinical motor and psychiatric impairments as assessed by the results of three motor scales scores referred to above: ICARS, CRST, and UPDRS-Motor. The predominance of intention tremor in the absence of gait ataxia or typical changes in cerebellar peduncles in these carriers led to speculation regarding the existence of modifying factors that might be accountable for neuroprotection in specific brain locations, such as the cerebellum. Notably, the data from an independent American sample presented by Hall, based on the low-symptomatic cohort of female PM carriers, were consistent with the above results by showing an isolated action tremor in some of the carriers who did not meet the criteria for FXTAS [143]. Further evidence for neural involvement in this sample was provided by a highly significant difference between female carriers and control noncarriers in the total score encompassing the three standard motor scales scores (FXTAS-RS). Overall, both Australian and American studies provided evidence for a diversity in the type and severity of neurological manifestations amongst carriers of PM alleles.

The data on general cognitive/executive functioning phenotypes in male and female PM carriers, though somewhat controversial, provide another example of a continuing neurodegenerative process across syndromic and non-syndromic categories of male and female carriers of PM alleles, which appear to be associated with the increasing size of the *FMR1* CGG repeat expansion. A string of early studies documented deficits in inhibitory control, working memory, planning, and attention in carrier males without FXTAS [224,239,240,241,242], with more recent work mapping these deficits to specific *FMR1* molecular genetic markers and alterations in brain regions important for executive functioning [243,244,245,246]. As reported at the International Premutation Conference, specific learning and attention problems resulting in daily function difficulties were a common feature of carrier females and were correlated with the size of the CGG repeats. In another study, daily function skills were predominantly impacted by dyscalculia (a learning disability in math), right and left disorientation, and attention deficits, such as ADHD [247]. The findings of the cognitive–executive deficits in female carriers were further supported by a series of case-control studies showing reduced performance in the areas of working memory, episodic memory, inhibition, attention, and language fluency/word retrieval [72,206,244,248,249,250,251,252,253]. Several studies reported CGG-dependent variation in some of these deficits, with the most severe impairments in women who carried midrange CGG sizes of about 80–110 repeats [254,255,256]. A growing number of cross-sectional reports, which have shown associations between older age and increased dysexecutive symptoms in PM women, have been suggestive of premature age-related decline [222,252,254,257,258]. Rare longitudinal research also demonstrated age-related decline of cognitive–executive skills in a subset of PM women with a family history of FXTAS, and identified a higher CGG repeat number as risk factors for a decline [259,260,261]. Segal et al. (2023) found an association between the number of CGG repeats and working memory among PM females [262]. Executive functions and phonological memory were assessed using the self-report questionnaire The Behavior Rating Inventory of Executive Function (BRIEF) and behavioral measures (nonword repetitions, forward and backward digit span). Female carriers reported less efficient executive functioning in the BRIEF questionnaire, which was correlated with the number of CGG repeats. However, these females did not report difficulties in reading or writing, and quite a few had advanced degrees and many years of education.

Both published studies and presentations given at the International Premutation Conference have shown that cognitive/executive impairments begin well before the age of onset of FXTAS. These impairments were represented by executive function deficits [240,251,263], memory problems [248,264,265], inhibition and attentional deficits [72,254,266], language dysfluencies [252], psychiatric problems [38,70,71,72,73,227], social–communication difficulties [69,212,255,267], sleep problems [268], subclinical or clinical motor problems [35,227,269], and MRI evidence of brain volumetric changes or functional changes [270,271]. Klusek and colleagues presented the results of their most recent, insightful study of subtle cognitive deficits in PM females. This study, including 90 PM carriers, assessed their performance on a cued recall paradigm that was especially sensitive to prodromal AD and mild cognitive impairment. This well-powered study demonstrated that PM females were much worse than matched controls on measures of proactive semantic interference and recovery, with very large effect sizes, despite the relatively young age of the sample (30–55 years, with a mean of 45); a significant association between the CGG repeat size and educational level was also recorded.

Several studies related the results of cognitive/executive assessments to the CGG repeat expansion size using linear or curvilinear models, with the latter showing that the middle range of the repeats (80–110) was generally related to the highest risk for the impairments, including executive function and memory difficulties, parenteral health, sleep quality, and maternal depressive symptoms [256,268,272].

A cross-sectional study of male PM carriers ranging in age from 18 to 69 years, compared with the noncarriers, showed that age was correlated with increasingly worse performance on measures of inhibitory control, working memory, and attention, two central components of executive functioning [239,273]. Importantly, the later studies, which included the size of the CGG expansion as a cofactor [240], led to the conclusion that that older age was, indeed, associated with decreasing executive function performance in male PM carriers, but only in those with more than 100 CGG repeats. A somewhat similar study concerning general cognition [274] administered a dementia rating scale in a double-blinded fashion to male PM carriers compared with intrafamilial controls. Using a cutoff for marked cognitive impairment, these authors determined that the penetrance of this impairment for mid to large (70–200) and small (55–69) CGG repeats was 33.3% and only 5.9%, respectively, compared to controls at 5.1%.

### 4.3. Do PM Cognitive and Motor Deficits Represent a Distinct Form of Neural Involvement, or Are They Prodromal to FXTAS?

The number of studies provided converging evidence of a dysexecutive pattern in carriers with and without FXTAS through tasks that tapped the effects of cognitive load on gait [275,276], fMRI studies [244,277], and a suite of structural MRI studies, including those that correlated the brain with cognitive measures [264,270,278,279,280,281,282]. But, the lack of any longitudinal studies prevented determination of a clear link that might establish cognitive changes as prodromal features of the later neurodegenerative disorder of FXTAS. An ongoing prospective study led by Hessl and Rivera, including neurological, neuropsychological, brain MRI, and molecular measures, focused on male carriers has begun to establish these links and identify risk and protective factors. Cognitive assessments using the Cambridge Automated Neuropsychological Test Battery (CANTAB) provided evidence that changes in visual working memory, inhibitory control, and planning progress were at a higher rate in PM male carriers than in a group of carefully matched controls, and that worsening inhibitory control and planning tracked the onset of FXTAS [15]. The most recent studies by this group, presented at the International Premutation Conference by Hessl et al., showed that these cognitive changes are reflected in the life experience of worsening executive dysfunction in these men. Using the Behavior Rating Inventory of Executive Function (BRIEF-A), a self-report scale of cognitive and behavioral regulation problems associated with executive function, the study reported longitudinal results on 66 PM men (40–78 years at baseline) and 31 matched controls assessed over two to five visits. Interestingly, despite the lack of any group differences on the BRIEF-A at baseline, the PM men showed greater decline in metacognition (self-initiation, working memory, organization, task monitoring) than controls across time. Conversion to FXTAS was related to age-related decline in behavioral regulation and metacognition [143]. Also, although PM men, on average, did not report more significant problems than controls at baseline, greater executive difficulties at baseline were associated with a higher likelihood of conversion to FXTAS at follow-up, a finding consistent with the earlier work of Kogan and Cornish (2010) [241].

Another notable example highlighting the dynamics of the effect of the PM allele on the observed wide clinical spectrum of neural involvement was presented by Loesch at the International Premutation Conference. She described the results of their follow-up study of the cohort of initially asymptomatic female PM carriers, where detailed neurological testing and scoring revealed the presence, and further progression, of subclinical motor and psychiatric (but not cognitive) impairments, as revealed by applying the three motor scales scores: ICARS, CRST, and UPDRS [238]. These initially subclinical impairments progressed within the original symptoms in a majority of these individuals, with only a small proportion converting into the syndromic FXTAS over the period of 10 years.

### 4.4. Major Psychiatric Issues (FXAND)

The elevated risks of psychiatric symptoms, reported in male and female carriers of PM alleles, are the most prominent illustration of the ubiquity of fragile X-associated changes occurring across non-FXTAS and FXTAS clinical categories [60,213,283]. The constellation of problems, meeting DSM-5 categorical criteria for a psychiatric disorder [284], has been termed FXAND.

FXAND includes anxiety, depression, insomnia, obsessive–compulsive disorder, chronic pain, and chronic fatigue [60]. Studies with females from FXS-affected families who are adult and who do not have FXTAS suggest that one or more of these problems occur in up to 50% of carriers across the lifespan and in both sexes [60,283]; however, results vary by the methodology, and there is minimal population-level data [73]. Using a broader approach, if anxiety and depression do not meet the DSM-5 criteria in severity for FXAND, then they fall under FXPAC, which uses the term “condition” instead of disorder. FXAND also includes ADHD and ASD-related social–personality, language, and neuropsychological features (described below in Section 4.4.4, Autism Spectrum Disorder and the Broad Autism Phenotype).

#### 4.4.1. Anxiety

Anxiety disorders have a current global prevalence of 7.3% (4.8–10.9%) [285]. In carriers of the PM, anxiety is also commonly experienced with a variety of clinical significance, and it is not only due to raising a child with FXS. Indeed, anxiety disorders may occur in carriers also in childhood and adolescence; thus, prior to having a child with FXS [36]. Cordeiro and colleagues (2015) found that carriers are at higher risk for anxiety disorders, such as generalized anxiety (GAD), specific phobia, social anxiety/phobia (SAD), and OCD, than in general population in a study of 35 individuals with a PM aged between 5 and 23 [36]. Overall, the first studies about emotional psychopathology in carriers have been performed with women. For instance, Franke et al. (1998) found higher rates of SAD and panic disorders (PDs) in mothers with a PM than in controls [286]. Several other investigations later showed greater frequencies of anxiety disorders in women with a PM than in the general population, whether the women with a PM had children or not [287,288,289]. Furthermore, Schneider et al. (2016) reported an elevated rate of self-reported OCD in female carriers compared to controls [290]. Additionally, anxiety disorders have been shown to co-occur with several other clinical conditions. For instance, Kenna et al. (2013), in a study performed with 41 mothers with a PM, found that 43% of them showed a comorbid history of anxiety and depression [291]. Furthermore, a study on 137 women with a PM presented at the International Premutation Conference by Kraan et al. evidenced that both social anxiety (~38%) and depression (~30%) occurrence were high in women with a PM, and that mental health issues were significantly associated with physical symptomatology, such as migraine and irritable bowel syndrome [143]. In males, anxiety disorders have also been reported. For instance, Bourgeois et al. (2011) underlined that male carriers with and without FXTAS are more likely to develop panic disorder and seasonal affective disorder than controls. Additionally, Santos et al. (2020) reported a case study of a 26-year-old man with a PM who presented with ID and seasonal affective disorder complicated by agoraphobia and selective mutism, which correlated with elevated symptoms of ASD [292].

Furthermore, neuropsychiatric issues have been reported to significantly reduce the quality of life (QoL) of people with a PM. For instance, Montanaro et al. presented at the International Premutation Conference data from a survey that they performed among Italian carriers with the aim to investigate the main symptoms, daily living challenges, and treatment priorities. The survey was completed by 51 individuals with a PM (49 females and 2 males), and results showed that anxiety represented the main area of concern for 78% of the respondents, and that both anxiety and depression were considered to have the greatest impact on the QoL of 82% of the respondents, who therefore considered the intervention for psychiatric symptoms a treatment priority [143].

Finally, while there is growing evidence of increased prevalence of anxiety disorders in adults with a PM, studies of medications are lacking. Both SSRIs and SNRIs are particularly deemed helpful for carriers with anxiety disorders [60]. As for nonpharmacological treatments in PM adults, only one study assessed the feasibility of an app-based mindfulness intervention in a small sample of women who are mothers of a child with FXS, showing that it was valid mostly in mothers with elevated SAD and stress. Additionally, physical exercise and a healthy diet style have been strongly suggested in carriers with a PM suffering from anxiety, with the aim to stimulate neurogenesis, improve mitochondrial function, and reduce stress [293]. Cognitive-behavioral therapy (CBT) is one of the most effective therapies used for anxiety disorders [294], but its efficacy has not yet been adequately evaluated in individuals with a PM. Finally, even though the combination of pharmacotherapy and psychotherapy seems to be the best option for the treatment of anxiety disorders under the umbrella of FXAND, controlled studies in carriers have not been carried out. Those and other studies focusing on the treatment of psychiatric problems in adults with PMs are then required.

#### 4.4.2. Depression

Adult PM carriers are at higher risk for psychiatric disorders, including depression (reviewed in [295]), which often requires treatment and falls under FXAND [60,295]. Some studies found a link between the number of CGGs and the prevalence [287], and to some extent, the severity of depression [296] of both sexes. PM carriers with more than 100 CGG repeat sizes had a significantly higher risk for depression [287]. A study of adults (119 males and 446 females) aged 18–50 found that the CGG length in males was marginally linked with depression (and negative affect) and only negative affect in female carriers, whereas there was no link with anxiety [296]. The rate of major depressive disorder (MDD) for reproductive-age females with a PM is high relative to the national average, with a higher-than-expected rate in the general population of their first MDD episodes occurring before the birth of a child with FXS [297]. The same study of 93 women also found that PM carriers with 70–100 CGG repeats had the greatest risk for DSM-defined MDD and a median age of onset of 27 years of age (in contrast to 15.5 years for AD). Similarly, another study added on to the growing literature on the curvilinear relationship between the CGG repeat length and depression [278], which found that women with the midrange (85 and 110) of CGG repeats had the highest prevalence of depressive symptoms compared to other PM female carriers. In contrast, the onset of depression in PM carriers with and without FXTAS was not associated with their number of CGG repeats, but found age and sex being relevant [298]. Namely, the study of 81 adult PM carriers (42% males) found a significantly higher median onset age of DSM-defined MDD in males (52 yo, and in those with FXTAS, 49.5 yo) than in the general population (32 yo), but not in those 58% females (34 yo); the latter is likely due to the intense stress of parenting children with FXS [295,297]. Importantly, as neurological issues in those males emerged significantly later than the MDD, that could serve as the prodrome to those who would develop FXTAS [298]. In terms of other mood disorders (e.g., dysthymia unipolar and bipolar), they are not reported to be higher in PM carriers compared with controls [67], although more data are needed to confirm those findings. Finally, there are a spectrum of possible confounding factors that could contribute to the occurrence of depression in PM carriers, including environmental, background genetic, and likely epigenetic factors [60,295]. Together, these data suggest that PM carriers are at risk for depression. While their phenotypic presentation can be subtle and of small effect size in some studies, which falls under the FXPAC, the aforementioned compelling data of the increased rate of DSM-defined MDD and need for treatment also clearly support the FXAND entity.

#### 4.4.3. Substance Abuse

Data are conflicting on substance abuse in individuals with a PM. A retrospective study of 24 women with a PM interviewed about their fathers with a PM provided initial evidence that males with a PM may have a higher incidence of alcohol abuse and dependence [299]. A later study assessed alcohol abuse in males with a PM and found that both males with a PM and family controls (males without a PM from the same family) both had higher rates of alcohol abuse compared to nonfamily controls, indicating a potential impact from the shared family environment [242]. In a study that included both males and females with a PM, PM carriers were not more likely to self-report a history of substance or alcohol abuse, though females (but not males) with a PM were more likely to report a personal history of alcohol consumption, defined as 50 or more alcohol drinks in their lifetime, compared to females without a PM [289]. More research is required to grasp substance and alcohol use among PM-allele carriers, its connection to mental health and comorbidities, and its potential impact on neurological outcomes affecting FXTAS onset [48,49].

#### 4.4.4. Autism Spectrum Disorder (ASD) and the Broad Autism Phenotype (BAP)

Building on extensive research showing significant overlap between ASD and FXS (e.g., [69,300,301,302,303]), various studies have noted increased ASD rates in children with the PM. However, these findings are constrained by clinic-based or small sample sizes. To ascertain the accurate prevalence of ASD in PM carriers, larger population cohort studies are essential, given the limited scope of current research utilizing clinic-based or small-sample approaches [37,39,41,42,56]. Although no studies have systematically studied the prevalence of ASD in adult PM carriers, evidence points towards elevated rates of the broad autism phenotype (BAP) in the PM. The BAP refers to a constellation of personality and language-related features that are qualitatively similar to the defining clinical features of ASD (e.g., rigid and socially reticent personality features, and differences in the use of language in social contexts, or pragmatics) that are typically more subtle and subclinical in expression [69,304,305]. The BAP occurs more frequently in first-degree relatives of autistic individuals and is believed to reflect genetic liability to ASD. Evidence suggests that PM carriers also demonstrate significantly elevated rates of BAP features [69,213,267,290,306,307]. For instance, using direct assessment tools to study BAP features in a group of over 150 women PM carriers, Maltman et al. (2021) reported that approximately half of the PM-carrier group displayed personality traits of the BAP. They further reported that the PM group exhibited more frequent pragmatic language violations, and a more dominant conversational style, compared to controls [213].

Differences in pragmatic language (i.e., the social use of language, such as conversational ability) are among the most frequently documented components of the BAP among women with the PM. The pragmatic language features of the PM phenotype were first described by Losh and colleagues (2012), who used detailed hand-coding to capture pragmatic language violations sampled from the conversation of 49 PM women, 89 mothers of autistic children, and 23 control mothers of typically developing children. PM women were more likely than controls to violate the social–pragmatic rules that govern conversation, such as abruptly changing the topic, going on tangents, or dominating the conversation [69]. The type and frequency of pragmatic language violations seen in PM women were similar to those seen in the BAP expressed in mothers of autistic children, supporting phenotypic overlap across the PM and the BAP that is consistent with a large body of literature implicating *FMR1* in autism liability (e.g., [308]). Additionally, children of PM-carrier women who exhibited higher rates of pragmatic violations showed elevated autism symptomatology. Given that *FMR1* is known to interact with a number of ASD-risk genes [308], such clustering of ASD-related features in a subset of families could have implications for the biological underpinnings of phenotypic variability in *FMR1*-related conditions.

A later report replicated the finding of elevated pragmatic language difficulties in an independent sample of PM women [309], and data presented at the International Premutation Conference also documented differences in speech-related features, including differences in speech rhythm, rate, and intonation, that contribute to pragmatics [143].

Further, differences in components of social cognition have been reported, including gaze behaviors contributing to social functioning. Social cognition is a critical skill contributing to pragmatics, where attention to social signals, and the ability to infer another’s thoughts and emotions, plays an important role in how language is deployed in social interactions. For instance, elevated pragmatic language violations among PM women appear related to differences in the use of eye gaze, which is reduced in some PM women during conversational interaction and does not normalize to the level of eye contact used by controls even following a “warm-up” period [267]. Several eye-tracking studies have also shown differences in attentional allocation to the eyes and faces in PM women that is linked mechanistically to differences in autonomic arousal, as measured by pupillary response and respiratory sinus arrhythmia [267,310]. Similarly, Maltman et al. also reported among PM-carrier women subtle differences in several dimensions of social cognition, including the ability to read complex thoughts and emotions from the eye region of the face, assigning complex emotional judgements to affective scenes, and faces varying in emotional valence [213]. Of note, these tasks have been linked with amygdala function, supporting prior findings tying amygdala dysfunction to social-information-processing difficulties in PM men [311].

While pragmatic language differences seen in PM women have typically been reported as subtle, accumulating evidence suggests an important clinical impact. PM women who experience more pragmatic difficulties report more depressive symptoms, loneliness, lower life satisfaction, and reduced family relationship quality [212]. Moreover, pragmatic language difficulties in PM-carrier mothers disrupt the synchrony of mother–child interactions [222], and are associated with poorer language skills and increased autism symptoms in children with FXS [69,312]. Thus, while pragmatic features in PM women may be “subtle,” their clinical relevance is not negligible, and knowledge of these features can be used to tailor family support services to optimize outcomes for both women and their children with FXS. A study presented at the International Premutation Conference by Friedman et al. reported that women with the PM have unique challenges with language that could not be attributed to working memory or attentional factors [143]. It will be critical in future research to delineate the presentation and developmental trajectory of the pragmatic-language phenotype of the PM during childhood, when efforts to intervene may have the largest effects. This is of particular importance given the high rates of anxiety in PM women and evidence that strong pragmatic language skills can buffer risk for developing anxiety and adjustment disorders during childhood [313,314].

### 4.5. Other FXPAC-Related Symptoms and Conditions

#### 4.5.1. Hypertension

Hypertension, or high blood pressure, is a common medical condition that affects approximately half of all adults in the United States [315]. Individuals with hypertension have an elevated risk of experiencing serious health problems, such as heart attacks, strokes, and kidney disease (NCCDPHP, 2021). Previous research suggests that PM carriers seem to be at a higher risk for developing hypertension relative to the general population, possibly due to diminished or absent levels of FMRP [316]. A study by Coffey et al. (2008) found that female PM carriers with FXTAS had a higher prevalence of hypertension relative to a group of age-matched controls; although hypertension was more common among females without FXTAS relative to females in the control group, this difference was not statistically significant [35]. Another study comparing hypertension in adult male PM carriers found similar findings to Coffey et al. (2008), such that adult male PM carriers with FXTAS were found to have a higher risk for hypertension relative to both male PM carriers without FXTAS and control participants [62]. These findings indicate that all PM carriers, particularly those with FXTAS, should undergo routine monitoring of hypertension and receive treatment if needed. Healthy lifestyle habits—including not smoking, eating well, exercise, and managing stress—can help prevent or manage hypertension, and should be encouraged for all PM carriers.

#### 4.5.2. Metabolic Syndrome

Metabolic syndrome (MetS) is a combination of conditions that increase heart disease, stroke, and diabetes risk. It results from overnutrition and sedentary lifestyles, manifesting as obesity, insulin resistance, dyslipidemia, and high blood pressure. MetS affects around 20–25% of adults, and its connection to PM carriers suggests a role of the *FMR1* gene in metabolism. Lifestyle and genetic factors may interact, elevating MetS risk in PM carriers. Research indicates elevated waist circumference, glucose, and lipid levels in PM carriers, with a higher MetS prevalence [317]. While more research is needed, monitoring metabolic health and early interventions are recommended for PM carriers to reduce cardiovascular risks.

#### 4.5.3. Chronic Fatigue

Chronic fatigue significantly affects the daily lives of individuals with PM, whether they have FXTAS or not [57,318]. However, carriers with FXTAS experience more severe fatigue compared to those without FXTAS and individuals in the control group [57]. Sleep apnea is frequently observed in patients with FXTAS and can contribute to the development of chronic fatigue [319]. There is also an indirect relationship between increased body mass index and fatigue due to its association with sleep apnea, diabetes, and coronary artery disease. Carriers without FXTAS exhibit intermediate levels of fatigue between FXTAS patients and controls. The study additionally indicates a relationship between fatigue and depression, highlighting that addressing depression can improve fatigue levels in patients [57,67].

The observed mitochondrial dysfunction in individuals with PM is likely linked to the fatigue experienced by them. Several studies have established a correlation between the severity of mitochondrial dysfunction and chronic fatigue in carriers [32,46].

#### 4.5.4. Chronic Pain and Fibromyalgia

Case reports and case series of chronic pain and fibromyalgia have been reported in PM-carrier women [320,321]. Subsequent studies have investigated an association between the gene and fibromyalgia. Fibromyalgia was increased in women with the PM with and without FXTAS compared to controls by Coffey et al. (2008) [35].

A case-control study was conducted to investigate neurological and endocrine phenotypes in women PM carriers compared to non-PM-carrier women [322]. A neurologist and endocrinologist, blinded to the gene status, examined each patient and reviewed a series of blood work related to endocrinologic diseases. Diagnostic criteria for fibromyalgia were performed, and participants were interviewed by the neurologist and had headaches classified according to the International Classification of Headache Disorders, Second Edition. Each woman was asked, regarding the presence of ‘central sensitivity syndromes’ based on standardized definitions, to include chronic fatigue syndrome, irritable bowel syndrome, temporomandibular disorder, myofascial pain syndrome, restless legs syndrome, periodic limb movements of sleep, multiple chemical sensitivity, primary dysmenorrheal, female urethral syndrome, and post-traumatic stress disorder [323]. In this study, the PM-carrier women were 54 ± 17 years, the CGG repeat (longest allele) was 91 ± 25, 96% white, and had 15+ years of education [322]. Women PM carriers did not have a higher rate of fibromyalgia or chronic fatigue syndrome compared to controls. PM-carrier women did have a significantly higher number of central sensitivity syndromes (3.4 vs. 0) compared to noncarrier women. The most common patient-reported diagnoses in the PM women were tension headaches, primary dysmenorrheal disorders, temporomandibular disorder, and interstitial cystitis. The study authors noted that the self-reported diagnoses were not completely concordant with physician diagnosis in the medical records, with physicians reporting irritable bowel syndrome and migraine headaches in addition to the others. The study authors in the discussion commented that the discordant results could be explained by difficulties in translating medical symptoms and signs into diagnoses during the clinic visit with the patient by the provider.

In the prior study, the NEO personality inventory was used to assess quantitative dimensions of normal personality traits [322]. In this cohort of participants, only the neuroticism profile was significantly different than controls (92 vs. 72, *p* = 0.02). The neuroticism domain contains items measuring anger, depression, self-consciousness, impulsiveness, anxiety, and vulnerability to stress. Each of the facets of the neuroticism domain independently contribute to negative affect and lower life satisfaction. Additionally, clinicians who see patients with high anxiety, hostility, self-consciousness, and depression can be confident that they have pervasive psychological distress. It is unclear if this profile is correlated with the results of increased central sensitization disorders or other findings in this study, but additional research may be needed to see if it may impact communication in the clinic room with the provider.

Larger screening studies have been conducted to look for *FMR1* PM alleles in women with fibromyalgia. A screening study for PM carriers in women with fibromyalgia (*n* = 353) found a higher-than-expected rate: 1/88 vs. 1/250 in the general population [324]. In a second screening study, 700 women with fibromyalgia based on the American College of Rheumatology 1990 criteria had DNA samples tested for the PM [325]. Only three PM carriers were identified (0.4%), and the authors concluded that the frequency of PM carriers is not higher than the general population. Screening studies in other chronic pain conditions have not been done.

In summary, chronic pain and fibromyalgia have been reported by PM-carrier women. However, case-control studies with blinded examiners and screening studies have not definitively confirmed these associations. They do suggest that additional work related to chronic pain, fibromyalgia and other central sensitivity syndromes in these women is warranted with a vigorous study design in the future.

#### 4.5.5. Sleep Problems

Sleep difficulties are usually observed in PM carriers even before the onset of their DSM-defined neuropsychiatric problems under the FXAND [60], especially problematic among adult carrier daughters of men with FXTAS. These women had significantly increased incidence of sleep problems compared to controls [326]. Sleep problems among PM carriers may be associated with sleep apnea [319], which may also be associated with opioid use in PM carriers [327]. Increased prevalence of sleep problems observed in PM carriers can be associated with some co-occurring conditions, such as ADHD and anxiety, in young PM carriers [41].

## 5. FXTAS Clinical and Protective Mechanisms

Not all individuals with the PM develop FXTAS. Having CGG repeats in the range from 50 to 60 may be protective for PM problems, and even FXTAS, because the *FMR1* mRNA levels are lower than a higher-end PM number; the higher the CGG repeat, the earlier the onset of FXTAS [11]. There are likely other genetic factors that can be protective against PM problems, and Hunter et al. (2012) documented those two polymorphisms in the corticotropin-releasing hormone type 1 receptor (CRHR1), which controls the release of ACTH, and subsequently cortisol levels, and influences the levels of anxiety and social phobia in women raising children with FXS [38]. Besides the genetic risks, there is evidence that stress in one’s lifestyle can lead to more frequent PM problems [256,328], and raising a child with FXS can be very stressful. We have also documented that other life events, such as surgery, particularly with isoflurane anesthesia [47], alcoholism, opioids, and other toxins [48,49], can be linked to the onset of FXTAS. We know that oxidative stress and mitochondrial dysfunction are seen in FXTAS and even in pre-FXTAS individuals compared to controls [31,32,190]. Brain volume changes and white-matter disease in carriers have been linked to decreases in mitochondrial mass and lowered ATP production [17]. So, treatments that improve these factors are likely to be helpful for FXTAS and possibly additional PM problems. We know that daily exercise can improve mitochondrial function, and a healthy diet and supplements such as sulforaphane [143,329] can also improve oxidative stress, but these interventions have not yet been studied thoroughly in the treatment of FXTAS [53]. Addressing excess stress, obesity, hypothyroidism, hypertension, diabetes, and other diseases, including psychiatric issues, through avoidance or early treatment, has the potential to impact brain health and potentially delay the onset of FXTAS.

Santos et al. reported at the International Premutation Conference on the results of the open-label trial of sulforaphane, an antioxidant and neuroprotective compound that protects neuronal mitochondrial function found in cruciferous vegetables, in 11 men and women with FXTAS. After 6 months of treatment, no significant motor improvements were noted; however, improvements were seen on measures of visual working memory and borderline significant improvement was seen on the Montreal Cognitive Assessment (MoCA). Although it is possible that improvements could be related to placebo or practice effects, it was noteworthy that a strong correlation was observed between the change in the FMRP level and improvement on the cognitive measures [143]. These studies have built nicely upon the foundations of prior work in the field and move us closer to finding effective targeted treatments for carriers with neuropsychiatric and neurological conditions.

Since most clinical research on FXTAS has focused on the symptoms, course, and correlates of this condition among PM carriers, much less is known about possible protective mechanisms—the factors that can reduce the likelihood of an FXTAS diagnosis or the progression of symptoms. Yet, some evidence points to the possibility of neuroprotection, drawing upon common patterns of both neuropathology and neuroprotection across neurodegenerative diseases.

As noted, the symptoms of FXTAS overlap with other neurodegenerative diseases, such as PD, Alzheimer’s, and others [330]. Some pathological mechanisms are common across different neurodegenerative diseases, and common treatment and protective mechanisms have been described. These include neuronal protection, repair, or regeneration, as well as modulation of neuroinflammation, bioenergetics, metabolism, and neurovascular interactions [331]. Examples of protective mechanisms shared across neurodegenerative diseases are the prevention (e.g., diet and exercise) and treatment (e.g., metformin and statins) of conditions known to increase the risk of cardiovascular disease, such as high blood pressure, diabetes, and hypercholesterinemia.

An additional shared mechanism across neurodegenerative diseases is higher education, which has been shown to reduce the genetic liability for age-related cognitive decline, including AD (e.g., [332] and PD [333]). Although not a primary focus of much PM research, many studies of the *FMR1* PM and FXTAS have incorporated years of higher education (i.e., postsecondary education) as a control variable in studies of the diagnosis of FXTAS or the development of FXTAS-type symptoms (including motor and cognitive functioning). The results are remarkably consistent—higher education appears to be a significant protective mechanism. For example, Storey et al. (2021) assessed the signs of neurological impairment in PM women and found that those with higher levels of education exhibited better motor and cognitive functioning [228]. Hartley et al. (2019) reported the results of an 8-day diary study of PM women who were mothers of adolescents and adults with FXS, and found that those who had a greater number of years of education had fewer daily physical health symptoms (including some that are present in FXTAS, such as fatigue, pain, muscle weakness, and dizziness) [334]. Klusek et al. (2020) found that educational attainment accounted for a significant portion of the variance in executive function deficits among PM women who had children with FXS [254]. In a study by Brega et al. (2009), 71% of PM carriers without FXTAS symptoms had 16 or more years of education, but only 43% of PM carriers with FXTAS symptoms had achieved a similar amount of schooling, a pattern also reported by Lozano et al. (2016) and Grigsby et al. (2016) [263,335,336].

However, in these studies, the effect of higher education for FXTAS-type symptoms was generally treated as a control variable. In contrast, at the International Premutation Conference, there were two presentations that focused specifically on higher-education effects. Neuroprotective effects of higher education were reported by Mailick, whereby PM women who did not attain a college degree had significantly more severe FXTAS-type symptoms than those who were college graduates, although the two groups were similar in age, CGG repeat number, household income, health behaviors, and general health problems [143]. Furthermore, symptoms manifested by those who did not attain a college degree worsened over the 9-year study period at a significantly faster rate than the college graduates. These results were published in Hong et al. (2022) [337]. Mailick further reported that, for women in the general population (i.e., not a clinical sample), years of postsecondary education interacted with the number of CGG repeats to predict later-life mortality. When mortality was assessed at age 80, women with CGG repeats in the *FMR1* gray zone and in the low-PM range who attended college had a longer survival than those who did not attend college, and they also had a longer survival compared with those who had fewer repeats. This research suggested a neuroprotective effect of higher education that was evident decades after college attendance.

Klusek et al. also reported at the International Premutation Conference that PM women who carried midsize CGG repeat lengths (approximately between 70 and 100 repeats) who had achieved a college degree had better cognitive function in midlife than those with midsize CGG repeat lengths who had not achieved a college degree. This gene-by-environment interaction was consistent with differential susceptibility, suggesting increased sensitivity to the protective effects of education associated with the midsize CGG repeat range [143]. A similar pattern of differential susceptibility at midsize CGGs has been reported in other studies of PM women and various phenotypes [272,328,334].

In this section of the International Premutation Conference paper, we examined the risk and protective mechanisms that may be related to whether a PM carrier develops FXTAS. Among the risk factors for developing FXTAS-type symptoms are older age and being male (although females also develop FXTAS). Additionally, genetic factors can affect the likelihood of a diagnosis of FXTAS or FXTAS-type symptoms. Having midrange or higher CGG repeats in the PM range increases the likelihood of the cognitive and motor symptoms of FXTAS, and for women, skewed X-inactivation plays a role. In addition, for many PM symptoms, a gene-X–environment interaction effect has been observed, whereby those who have midsize CGG repeats appear to be more sensitive to both positive and negative aspects of the environment than those who have higher or lower repeats within the PM range. For the motor phenotype, the phenotypic presentation at symptom onset is clinically important, with those who have tremor as a first sign having milder impairment than those with ataxia as well as tremor when first diagnosed. The importance of diet and exercise as protective factors has been recognized in clinical research. Additionally reported at the International Premutation Conference was the protective factor of a college education, substantially reducing the risk and severity of FXTAS symptoms. These protective factors suggest strategies for reducing the age of onset and the severity of FXTAS-type symptoms that can be helpful in addition to medical treatments. The protective effects of higher education and adhering to a healthy lifestyle point to potential socioeconomic factors that differentiate the healthier members of the PM population from those who are more symptomatic [143,338].

## 6. Reproductive and Health Implications in Women Who Carry the PM

### 6.1. Fragile X-Associated Primary Ovarian Insufficiency (FXPOI)

FXPOI affects women in reproductive age, leading to irregular menstrual cycles and infertility. It is defined as at least four months of unpredictable or absent menstrual periods and two serum-follicle-stimulating hormone (FSH) levels in the menopausal range at one month apart [339] in females of less than 40 years. Infertility is defined by the inability to conceive after 12 months of unprotected intercourse. However, for women aged 35 and older, the inability to conceive after 6 months is generally considered infertility. FXPOI is caused by the PM, and it is a leading cause of genetic infertility, affecting about 1% of women [340]. Recent studies have also suggested a possible association between POI and polycystic ovary syndrome (PCOS), a common endocrine disorder that also affects women in reproductive age [341]. Reduced fertility is the most immediate and significant consequence of diminished ovarian function. Other consequences of POI, primarily related to early estrogen deficiency, affect quality of life and overall health and mortality (e.g., reviewed in [342,343]). POI, in general, is known to have significant negative impacts on a woman’s health. Following the description of FXPOI in women with the PM in 1999 [3], medical comorbidities related to FXPOI, such as osteoporosis, were identified [344,345]. Other comorbidities include depression, anxiety, and other neuropsychological problems, as well as reduced bone mineral density and an increased risk of cardiovascular disease [346]. Management for women with the PM includes genetic counseling regarding genetic risk for offspring and reproductive options, including fertility preservation with egg retrieval, in vitro fertilization, and preimplantation genetic testing [347].

### 6.2. Medications to Treat FXAND in FXPOI

The most common conditions under the FXAND umbrella are anxiety and depression in adults with the PM [70], including females with FXPOI. Indeed, among women with FXPOI, there is a high rate of an earlier age at onset of anxiety, and within a FXPOI–mental-health-problems cluster, a higher proportion of those women have a child with FXS [348]. An early diagnosis and treatment of the comorbid anxiety and depression in FXPOI is being recommended [53], and both SSRIs (i.e., sertraline or escitalopram) and SNRIs (i.e., duloxetine or venlafaxine) are suggested [349,350]. If pain symptoms are a part of FXAND, then the use of an SNRI initially, such as duloxetine or venlafaxine, is recommended to treat both the pain and the depression/anxiety [351]. The SNRIs may also be more beneficial when ADHD symptoms are a part of FXAND in adulthood [352]. Duloxetine is also beneficial in treating premenstrual dysphoric disorder (PMDD) in women [353], and even more evidence-based data exist for use of SSRIs in PMDD [354,355].

### 6.3. Psychotherapy to Treat FXAND in FXPOI

The treatment of anxiety and depression should encompass psychotherapy, an evidence-based practice in psychiatry [356,357,358,359]. Cognitive-behavioral therapy (CBT), a structured talk therapy, stands as a gold-standard for treating depression and anxiety disorders in adults [360]. Meta-analyses support CBT’s effectiveness against perinatal and postnatal depression, as well as PMDD [361,362]. CBT, conducted by trained nonphysician professionals, can extend to addressing comorbid medical and neurological conditions [356]. Combining CBT with medication yields enhanced efficacy, including benefits for ADHD [363]. The current trend in CBT incorporates mindfulness and dialectical behavioral therapy [364]. However, implementation hurdles include training, access to mental health recommendations, and time constraints [365,366]. Notably, many depression patients receive primary care provider treatment due to access challenges [367].

General suggestions for FXAND: in addition to regular exercise [293,368], avoidance of toxins in the environment, including excessive use of alcohol or opioids, is recommended to patients who carry a PM [48,49,60].

### 6.4. Early Diagnosis and Carrier Screening

Early diagnosis and management of POI are crucial to prevent or mitigate adverse health outcomes. Therefore, carrier screening can significantly improve early diagnosis and has been implemented in some countries. Carrier screening can be also improved with the inclusion of AGG-interruption analysis. In a recent study, reported at the International Premutation Conference, Archibald and colleagues described that, out of 46 females with the PM identified from 2020 to 2022, 37 (80.4%) had a small PM allele. Following AGG-interruption analysis, 32 of these 37 females (86.5%) were reported to have a low reproductive risk for FXS, while the remaining 5 (13.5%) were reported to have an increased reproductive risk for FXS. This reduced the number of clinically actionable results by 69.6%. Moreover, the reproductive outcomes indicated that women with a small stable PM were reassured of their low reproductive risk. This report highlighted the improvement of the clinical utility of *FMR1*-carrier screening; by reducing the need of prenatal diagnosis and/or PGT-M for FXS, genetic-counseling resources can be focused on supporting those with a PM with a higher risk of having children with the full mutation [143].

A study by Allen and colleagues described the qualitative healthcare experiences of women who carry a PM. This presentation summarized two projects that were done by genetic-counseling students at Emory University in Atlanta, Georgia. In the first project, Bonnie McKinnon Poteet conducted 24 interviews with women who had a diagnosis of FXPOI to identify barriers and facilitators for their diagnosis. Overall, common themes among women included hopes for broader physician awareness of FXPOI, more clear guidelines for treatment, and proper fertility options prior to diagnosis to expand their reproductive options. Further, the women also spoke of the need for centralized care or an “FXPOI navigator” to help them before, during, and particularly after they received their diagnosis. Because this initial study was on a predominately white population, this project was followed up by a second project in African-American women by Andy King to identify what the healthcare experiences are for African-American women who carry a PM. After interviewing eight women, the identified themes from these interviews included concerns about healthcare provider dismissal, isolating the lack of support from family members around their diagnosis, and a high incidence of anxiety and depression. Participants consistently reported multiple caretaking roles, including in their employment, and providing strength and support to others. There are several areas for improvement for care for PM women overall based on these results, including more centralized care, improved clinical care, and increased support [143].

### 6.5. Future Directions

Facilitating communication among patients, community health providers, and researchers is crucial to improving patient care and advancing research. The advent of telemedicine and mobile-health technologies has improved access to care and research participation, and these methodologies can be used for a national effort on the natural history of this condition. A national natural-history study on FXPOI is needed to identify potential biomarkers and treatment options. In fact, recent work by Shelly and colleagues, using untargeted metabolomic profiling (LC/MS/MS) in human plasma, analyzed two cohorts of women, including the largest test cohort of PM women to date (40 FXPOI cases, 34 PM controls) and a validation cohort (22 FXPOI cases, 58 PM controls). They found altered levels of omega-6 fatty acid (n-6 FA) and arachidonic acid (AA) metabolites in FXPOI when compared to female PM carriers without FXPOI in both cohorts. The downstream metabolic markers of arachidonate, including prostaglandins and proinflammatory metabolites, were also changed in women with FXPOI, specifically. They also confirmed these changes when they accounted for the menopausal status of the controls. This revealed that individual metabolites in the n-6 FA pathway were higher when all individuals were postmenopausal, but FXPOI cases showed decreased levels when controls were premenopausal. Shelly et al. previously published that downstream products of AA were perturbed in a mouse model of the PM [369], and these alterations were linked to ovulation. By comparing metabolite pathways changed in patient plasma to differential gene expression in PBMCs of a preliminary patient cohort, the group explored whether this connection existed in the human FXPOI population. They found that transcripts related to fatty-acid processing and the generation of prostaglandin precursors were altered and were explicitly related to ovulation through gene set enrichment analysis. Further studies are necessary to identify the molecular mechanism that leads to ovarian infertility, as well as biomarkers that can be used to identify early states of ovarian insufficiency. This work suggests that markers, such AA, generally, and prostaglandins and prostaglandin synthases, specifically, may become useful biomarkers to identify individuals with FXPOI. It also supports growing evidence that ovulation may be the key period of follicle development [370].

In addition to these findings on metabolomics, Allen and colleagues described genetic and environmental data from the same cohort of women that have been collected at Emory University. In this work, they described the confirmation of previous studies where a nonlinear relationship with risk for FXPOI and the *FMR1* CGG repeat size are seen, with women with the midrange of PM CGG repeats being at the highest risk for FXPOI [4,371,372,373,374]. Based on the transcriptome-wide association study (TWAS) that was carried out on 106 PM women with FXPOI (menopause < age 35) and 101 PM controls (menopause ≥ age 50), five genes were found to be significantly associated with risk for FXPOI: TCAM1P, PRR29, CEP95, ACE, and FTSJ3. These risk genes are all known to be associated with age at menopause or hormone levels. The authors are currently investigating environmental risk factors, such as residential or occupational history, in addition to smoking history, which is known to affect age at menopause in their study population.

Although only a few presentations were given at the recent International Premutation Conference that specifically focused on the reproductive implications of the PM, it is noteworthy that the talks were able to cover such a broad spectrum of topics. First, the utility of using the size of the CGG repeat and AGG interruptions to determine the risk more accurately for having a child with FXS and reduced the need for prenatal diagnosis or PGT-M for many women. Second, the healthcare experiences as described by women with a PM through qualitative interviews identified many areas where we as researchers and clinicians can better serve this population. From the basic science work, we are starting to identify pathways and genes that are affected in women with FXPOI compared to women who have a more typical age at menopause. These molecular clues will hopefully lead us to a full understanding of the pathogenesis of FXPOI and direct us toward therapeutic options in the future.

## 7. Neuroimaging Findings in FXTAS

### 7.1. Structural Brain Differences Associated with FXTAS

Decades of research on fragile X-PM carriers have revealed many structural MRI features of FXTAS. Early MRI work reported generalized brain atrophy, corpus callosum thinning, widespread white-matter disease and loss of integrity, and enlarged ventricles [244,277,375,376]. The most prominent brain regions showing consistent structural changes in FXTAS across different studies are the cerebellum and brainstem [244,271,376,377]. In a large cross-sectional study of 142 male controls and 181 male PM carriers with and without FXTAS (age 8–81 years), both cerebellar and brainstem volumes showed abnormal age-related changes in carriers without FXTAS and atrophy in carriers with FXTAS compared with controls [270]. These findings suggest that cerebellar and brainstem regions are likely affected during both neurodevelopment and FXTAS-associated neurodegeneration. Quadratic relationships with the CGG repeat length were revealed as well, indicating structural brain differences affecting the cerebellum and brainstem may disproportionately impact PM carriers with midrange CGG expansions [244,270]. Consistent with these findings, separate studies have documented reduced cerebellar and brainstem volumes in carriers without FXTAS, suggesting structural differences may precede the onset of FXTAS or manifest in PM carriers independent of disease status [244,376,378]. Support for the hypothesis that cerebellar degeneration may serve as a prodromal marker of FXTAS decline comes from research showing that MRI measures of the cerebellum may be useful for predicting the non-FXTAS to FXTAS conversion. In a longitudinal study, the MCP width was reduced in male PM carriers who did not show FXTAS symptoms initially but developed FXTAS symptoms during follow-up visits compared with PM carriers who remained symptom-free [379]. Structural damage in the cerebellum may play an important role in FXTAS symptomatology. Correlations between cerebellar atrophy with FXTAS severity [244,270], gait impairment [265,380], and slowed step initiation [381] have been demonstrated, and microstructural white-matter disease in the MCP and SCP appears to be related to executive dysfunction and reduced dexterity in male PM carriers [148,382].

Other prominent brain regions involved in FXTAS include the thalamus and basal ganglia, a group of subcortical nuclei interconnected with the cerebellum to form cortico-basal ganglia–cerebellar networks important for motor, executive, and emotional processing [383,384]. The thalamus and select nuclei in the basal ganglia, namely the caudate, putamen, and globus pallidus, show atrophy and diffusion-weighted signal loss consistent with iron accumulation in male patients with FXTAS [380,382]. Thalamic volume loss has been linked to increased gait variability [380], and associations between caudate atrophy and slowed information processing speed also have been demonstrated [278]. In addition to T2-hyperintensities in the MCP and corpus callosum that are used as radiologic criteria for FXTAS diagnosis [231,322,385,386], T2-hyperintensities in the globus pallidus (the pallidal sign) recently have been reported in an MRI study of 257 male controls and PM carriers with and without FXTAS (age > 45) [281]. While 52% of PM carriers showed the MCP sign versus 0% of healthy controls, 25% of PM carriers and 13.4% of controls showed the pallidal sign, and 16% of PM carriers versus 0% of controls showed both the MCP and pallidal signs. Importantly, the presence of the MCP sign was associated with action tremor, cerebellar ataxia, and executive dysfunction, and the presence of both signs was associated with more severe executive dysfunction [281]. Except for one study containing only patients with FXTAS [231], the presence of T2-hyperintensities in specific brain regions were evaluated by raters blinded to the *FMR1* gene status of the participants [281,322,385,386]. The hippocampus and associated fiber tracts also show structural alterations in both non-FXTAS and FXTAS [244,387] and are related to more severe paranoid ideation and memory impairment in male PM carriers [264,388,389]. The corpus callosum represents an additional white matter fiber tract that may be linked to declines in executive abilities and processing speed in FXTAS [148,390].

Understanding the pathophysiological mechanisms underlying brain structural changes in FXTAS is vital for the discovery of effective therapeutics. White-matter hyperintensities (WMHs), indicating brain structural damage, are important MRI features of FXTAS that have also shown associations with motor and cognitive deficits in FXTAS [391]. The severity of WMHs is associated with multiple *FMR1* molecular markers (i.e., CGG repeat length and mRNA level), peripheral measures of mitochondrial bioenergetics, and the activity of a cellular stress marker, the enzyme AMP-activated protein kinase [17,180,392]. In a recent study applying artificial neural network analysis [393], the combination of both the mitochondrial bioenergetics and MRI measures of WMHs and whole brain volumes was useful for classifying the FXTAS stage in 127 male and female PM carriers with and without FXTAS. In addition, a longitudinal study investigated the effect of ventricular expansion on brain deformation using a periventricular white-matter structure, the corpus callosum, and a nonperiventricular nucleus, the putamen [394]. The study revealed 48.6% individuals with FXTAS met Evan’s index criterion (>0.3) for normal pressure hydrocephalus, a progressive ventricular expansion from the fourth to the third, and then the lateral ventricles, and a deleterious cycle between ventricular expansion and atrophy and deformation in the corpus callosum and the putamen [394]. These studies indicate that targeting bioenergetics, white-matter disease, and ventricular expansion may prove effective for delaying FXTAS progression.

### 7.2. Functional Brain Differences Associated with FXTAS

Despite accumulating knowledge of structural brain differences associated with FXTAS, the understanding of the functional brain changes that underpin clinical decline remains limited. This knowledge gap critically slows treatment development because key brain targets for new therapeutics have not been identified, and objective readouts sensitive to target engagement and treatment outcome in clinical trials are not yet available. Clarification of functional brain changes has accelerated the development and validation of targeted therapeutics in separate diseases of aging (e.g., Parkinson’s and AD), suggesting greater research attention on functional brain changes that track with clinical decline in FXTAS is needed [395,396].

Initial quantitative electroencephalography (EEG) and event-related potentials (ERPs) studies of FXTAS provide evidence that objective measures of brain functions in FXTAS will be important for advancing more effective treatment options. Yang and colleagues first identified atypical ERPs in FXTAS patients relative to healthy controls during tests of attention and working memory (2013), as well as verbal learning and memory (2014a). To assess neurophysiological processes associated with executive dysfunction in FXTAS, Yang et al. (2013) presented individuals with an auditory oddball task in which they were instructed to press a button in response to “infrequent” tones presented on 25% of trials, and to count “frequent” tones that were presented on 75% of trials. This experiment reliably elicits P200 components during target tones associated with selective attention, and prominent P300 components over the parietal cortex during the processing of oddball or infrequent targets. During the auditory oddball task, FXTAS patients showed attenuated P200 and P300 amplitudes relative to healthy controls, suggesting these components could be useful outcomes in clinical trials focused on mitigating executive dysfunction in FXTAS. This hypothesis is supported by a subsequent clinical trial of memantine, an NMDA receptor antagonist approved for the treatment of AD, in which the same investigative team showed increases in P200 amplitudes among FXTAS participants relative to FXTAS patients randomly assigned to the placebo [282,397]. The same team documented a similar promise for EEG/ERP measures of verbal encoding and memory. During a semantic judgment task, in which healthy controls demonstrated a neural “repetition effect” in which the amplitude of the N400 component decreased over multiple trials of the same word, FXTAS participants showed a limited change in the N400 amplitudes across word repetitions [398]. In a follow-up trial of memantine, both cued-recall memory and N400 repetition amplitude differences improved, whereas the placebo was associated with a worsening of each outcome in the FXTAS patients [399]. In the context of a prior trial of memantine in FXTAS, in which participants showed no significant changes on neuropsychological measures of executive function, verbal learning/memory, or working memory [400], the EEG/ERP studies underscore the significant potential of the quantitative assessments of brain function to expedite drug-discovery efforts in the context of FXTAS.

While EEG/ERP strategies offer highly temporally precise and scalable approaches for testing neurophysiological changes in FXTAS, they are limited in their potential to elucidate functional brain differences involving subcortical and cerebellar/brainstem circuits that have been implicated in structural and neuropathological studies of FXTAS. Functional magnetic resonance imaging (fMRI) approaches, particularly task-based fMRI methods that have been shown to be more strongly associated with cognitive and behavioral traits relative to task-free fMRI [401], offer a greater spatial resolution than EEG/ERP, and are capable of measuring changes in the activation and functional connectivity across subcortical and posterior fossa networks. Despite these advantages, few fMRI studies of FXTAS patients have been conducted, and only two known studies have examined the primary behavioral features of FXTAS—motor impairment. Brown et al. (2018) first documented the reduced functional activation of cerebellar motor regions, including lobules V and VI, and the hippocampus during a finger-tapping test [402]. McKinney et al. (2020) subsequently examined the visuomotor behavior in a sample of aging PM carriers (45–74 years) using an fMRI test of precision gripping. Briefly, participants held a precision force transducer while viewing two horizontal bars, including a “force bar” that moved upwards with increased force, and a “target bar” positioned at a fixed location above the force bar. Participants were instructed to press so that the force bar reached the level of the target bar, and to hold their force level as steadily as possible [403]. Using a similar test outside of the MRI environment, Park et al. (2019) and McKinney et al. (2019) each found that aging PM carriers showed greater force variability than age- and sex-matched healthy controls [404,405]. During the fMRI, McKinney et al. (2020) documented that aging PM carriers, including four with possible, probable, or definite FXTAS, showed reduced functional connectivity between ipsilateral cerebellar Crus I and extrastriate cortex during pressing, which was associated with both the increased CGG repeat length and greater force variability. These findings suggest that PM effects on motor function involve atypical communication between visual-processing cortical circuits and cerebellar circuits involved in translating sensory-feedback-error information into corrective motor commands [403]. These results are consistent with the findings of the degeneration of white-matter microstructural integrity affecting the cortical and cerebellar/brainstem pathways [394]. Promising fMRI studies probe the motor network function for potential biomarkers tracking degeneration in FXTAS. Longitudinal studies are vital, comparing changes in patients with FXTAS and asymptomatic PM carriers. These studies are needed to establish biomarker utility in distinguishing degeneration from aging processes in the PM carriers. They are also important to track target engagement and therapeutic outcomes effectively.

The diverse range of cognitive and behavioral abilities affected by FXTAS suggests that an array of functional imaging strategies will be important for developing disease-modifying therapies. Multiple fMRI studies of FXTAS patients have begun to identify functional brain differences associated with executive impairments [244], the encoding of new information [277], associative recall [246], magnitude estimation [245], and social–emotional processing [311,406]. Collectively, these results suggest FXTAS is characterized by generalized attenuation of functional activation during cognitive and social behaviors. Multiple studies also document reduced functional connectivity of widely distributed cortical–cortical and cortical–cerebellar networks, consistent with radiological, quantitative structural, and histopathological studies showing prominent degeneration of white-matter pathways involved in long-term network communication [277,403]. Integrating measurements of changes in the white-matter microstructure and task-dependent functional connectivity may offer a powerful approach for clarifying key neurodegenerative processes contributing to clinical declines in FXTAS, identifying targets for therapeutic development, and developing functional biomarkers useful for assessing the efficacy of new treatments.

## 8. The Neuropathology of FXTAS

FXTAS is characterized by the presence of intranuclear inclusions in neurons and astrocytes. Inclusion burden is positively correlated with the *FMR1* CGG repeat length [11]. Inclusions are larger and more prevalent in astrocytes and have been observed in several brain regions, including the hippocampal formation (most numerous), cortex, thalamus, basal ganglia, substantia nigra, inferior olivary, dentate nuclei, pons, and cerebellum [11,407]. They have also been identified in the endothelial cells of small vessels [20], ependymal and subependymal cells, choroid plexus [407], cranial nerves, spinal cord, and in other non-nervous tissues, including the heart, pancreas, intestine, kidney, and testis [66]. On a hematoxylin–eosin stain (H&E), inclusions are discrete, hyaline-appearing, eosinophilic, and have a round/ovoid body (Figure 3a–c) [18]. They measure 2–5 μm in diameter and are almost unanimously single, except in Purkinje cells (PCs), which sometimes present with two inclusions that are known as twin inclusions [408]. Inclusions are periodic acid–Schiff (PAS), Tau-negative, and ubiquitin-positive. A proteomic study of the FXTAS inclusions found several proteins of interest, including the small ubiquitin-like modifier (SUMO2) and p62/sequestosome-1 (p62/SQSTM1), both involved with the ubiquitin–proteosome system. Other remarkable proteins involved with protein turnover, DNA-damage repair, and RNA-binding were also found [134]. Repeat associated non-AUG translation occurs in FXTAS, resulting in the production of toxic peptides, including the glycine-rich FMRpolyG. FMRpolyG-positive inclusions are also found in the FXTAS brain [409].

Neuropathological and radiological studies have demonstrated changes in the FXTAS brain indicative of widespread neurodegeneration and inflammation. Neurodegeneration is manifested by regional reductions in brain volume, white-matter (WM) disease, iron deposition (Figure 3f), and microbleeds. WM disease is particularly severe, seen in the cortical white matter, corpus callosum, and cerebellum (Figure 3d), and is accompanied by regional atrophy. FXTAS WM disease includes spongiosis, axonal degeneration, myelin loss, and, infrequently, axonal torpedoes. The middle cerebellar peduncles, which can present with an increased T2 signal intensity on magnetic resonance imaging (MRI) scans in individuals with FXTAS, often show myelin pallor on luxol fast blue/periodic acid–Schiff stain (LFB-PAS). Gray matter atrophy also occurs, which is particularly severe in the cerebellum, pons, and striatum, and is often associated with ventriculomegaly (Figure 3e) [394].

Considering the characteristic motor symptoms of FXTAS, cerebellar involvement is prominent. Observations from cerebellar tissue include a remarkable dropout of Purkinje cells, Bergmann gliosis, and Purkinje axonal torpedoes [11]. In one study, iron measurements were collected from 12 FXTAS and 13 control in the cerebellar cortical and dentate regions. However, the number of iron deposits in the cerebellum only increased in a subset of FXTAS cases; thus, making it ineffective as a hallmark of FXTAS pathogenesis [410]. Iron localization using Perl’s method and iron-binding protein immunostaining was also assessed in the putamen from nine FXTAS and nine control cases. There was increased iron deposition in neurons and glial cells in the putamen, and a generalized decrease in the amount of the iron-binding proteins transferrin and ceruloplasmin, and decreased number of neurons and glial cells that contained ceruloplasmin. However, there were increased levels of iron, transferrin, and ceruloplasmin in the microglial cells, indicating an attempt by the immune system to remove the excess iron. Overall, there was a deficit in proteins that eliminate extra iron from the cells with a concomitant increase in the deposit of cellular iron in the putamen in FXTAS [411]. In addition, postmortem choroid plexus from FXTAS and control subjects found that iron accumulated in the stroma, transferrin levels were decreased in the epithelial cells, transferrin receptor 1 distribution was shifted from the basolateral membrane to a predominantly intracellular location (FXTAS), and ferroportin and ceruloplasmin were decreased within the epithelial cells [19].

It has been suggested that FXTAS can be a small vessel disease. A study in cortical and cerebellar tissue from 15 FXTAS and 15 control cases found intranuclear inclusions in the endothelial cells of capillaries (Figure 3m,n) and an increased number of cerebral microbleeds (Figure 3l) (predominantly in the WM), both indicators of cerebrovascular dysfunction. In addition, an association between the number of capillaries that contained pathologic amounts of amyloid β, consistent with mild-to-moderate cerebral amyloid angiopathy in the cerebral cortex, and the rate of FXTAS progression, was also observed [20]. A postmortem MRI study reported higher ratings of T2-hyperintensities (indicating cerebral small vessel disease) in the cerebellum, globus pallidus, and frontoparietal WM, consistent with findings in histology [412]. Characteristic hypertensive pathologies, such as arterial wall hyalinosis and widened perivascular space around the vessels, were mild in nearly all of the FXTAS cases, except for one case due to attributable hypertensive cardiovascular disease [18].

The neuroinflammatory profile of FXTAS includes the activation of both microglia and astrocytes, and elevations in the brain levels of specific cytokines. Using Iba1 and CD68 antibodies to label the microglia, the number and state of the activation of microglial cells in the putamen of 13 FXTAS and 9 control cases were examined. Nearly half of the FXTAS cases (6 of 13) presented with senescent microglial cells, characterized by dystrophic and fragmented morphology. The remaining cases (7 of 13) showed a robust increase in microglial activation (Figure 3g,h) compared to controls. In another study, striatal and cerebellar tissue from 12 FXTAS patients and 12 matched controls was immunostained for GFAP. The FXTAS cases showed severe reactive gliosis in both gray matter (GM) and WM (Figure 3i,k). Reactive astrocytes had gemistocytic cell bodies, intense GFAP staining, and process blebbing (Figure 3j; please note that gemistocytes are glial cells that are characterized by billowing, eosinophilic cytoplasm, and a peripherally positioned flattened nucleus). A substantial reduction in astrocyte numbers (30–40%) was found exclusively in the WM of the putamen and cerebellum. Additionally, numerous reactive astrocytes were positive for cleaved caspase-3, suggesting that apoptosis-mediated degeneration is responsible for the reduced astrocyte number. Neuroinflammation is largely regulated by both astrocytes and microglia within the brain, which utilize cytokines to coordinate the neuroinflammatory response. Microglia are the primary source of cytokines in the nervous system, and synthesis and secretion are upregulated when activated. A recent study characterized cytokine alterations in the FXTAS brain using a commercially available ELISA panel. They found a large significant increase in the cytokines interleukin-12 (IL-12) and tumor necrosis factor alpha (TNFα), major mediators of inflammatory and regulators of immune responses [413]. There were large, but nonsignificant, increases in the levels of IL-2, IL-8, and IL-10 in FXTAS. The cytokines IL-1α, IL-1β, IL-4 IL-6, IL-17α, IFNγ, and GM-CSF were not different between the FXTAS and control groups. TNFα and IL-12 are both implicated in the pathogenesis of multiple sclerosis, another neurodegenerative disorder that predominantly consists of WM disease [413].

Recent studies showed the frequent coexistence of FXTAS with other neurodegenerative disorders. About 50% of FXTAS patients develop dementia [22], and it is common to find classic parkinsonian features, including bradykinesia and muscle rigidity, during clinical evaluations [414]. A systematic review of medical histories from 70 postmortem brains with FXTAS found that 23% were clinically diagnosed with dementia. In a single postmortem brain study to date in females with FXTAS, half of them were additionally diagnosed with dementia, while AD pathology was found in 75% of the cases [154]. Nevertheless, female gender is a known risk factor for AD, and although the prevalence of FXTAS-AD is unknown, it is not expected to be the main etiology for cognitive impairment in FXTAS, since the pattern of cognitive deficits in FXTAS is different from that of AD [22]. However, limited data highlight the faster progression of motor and cognitive abilities, and faster than normally seen brain atrophy in individuals clinically diagnosed with FXTAS and AD [13,20]. In an analysis of forty FXTAS postmortem cases, five were clinically diagnosed with idiopathic Parkinson’s disease (PD) and two with atypical parkinsonian syndrome. After pathological examination, all cases had dopaminergic neuronal loss; however, only two of seven presented Lewy bodies in the substantia nigra. Based on these findings, approximately 3–5% of FXTAS cases were present with concomitant PD [21]. Other comorbidities include two cases of FXTAS with the inclusion body myositis [415], progressive supranuclear palsy [416,417], and one case of the Prader–Willi phenotype [415].

## 9. FXTAS Treatment

While ongoing studies seek to clarify the pathophysiology and neuropathology of FXTAS, as well as the development of meaningful biomarkers for the onset and progression of FXTAS [418], there are currently no targeted treatments for FXTAS that will reverse the core neuropathology [419]. Instead, current clinical management is symptom-directed [12] and includes the off-label use of medications for movement disorders (i.e., tremor, ataxia, parkinsonism) [105,420,421,422,423] and other neurological disorders (i.e., cognitive decline, memory loss, nerve pain/chronic pain [34], and FXAND problems [60,116]).

### 9.1. Treatment Trials Specific to FXTAS

Numerous smaller open-label studies have been conducted. For example, a one-year, double-blind, randomized placebo-controlled trial enrolled 94 patients with FXTAS and used memantine, a noncompetitive NMDA antagonist approved for AD. While motor and neuropsychological outcomes showed no significant difference, memantine displayed benefits in cued memory recall and attention/focus (auditory odd-ball paradigm) compared to the placebo, implying a cognitive influence in FXTAS [397,399,400].

Allopregnanolone is a naturally occurring neurosteroid that is a GABA agonist that stimulates neurogenesis in the hippocampus. Allopregnanolone blood levels are high during pregnancy, but the levels decrease after parturition. Allopregnanolone is currently FDA-approved for IV treatment in postpartum depression. In hippocampal neurons cultured from PM mice, allopregnanolone treatment normalized the enhanced spike-burst patterns, suggesting possible benefits for patients [55]. Therefore, an open-label 3-month study of a weekly IV allopregnanolone treatment (2 to 6 mg per dose infused over 30 min) was carried out in six patients with FXTAS [424]. All patients tolerated the treatment well without significant side effects. Those patients with relatively normal-sized hippocampi and corpus callosum at baseline on MRI demonstrated improvement in neurocognitive testing, particularly executive function and memory testing, after 3 months of treatment. One patient demonstrated resolution of his neuropathy and improved ataxia on allopregnanolone, but tremor severity did not improve. Molecular changes were seen in the GABA metabolism pathway, oxidative stress measures, and mitochondrial-related outcomes [191]. These effects in a small open-label study suggest that further studies of allopregnanolone are warranted [425]. Importantly, an oral allopregnanolone preparation has been developed and will hopefully be studied in patients with FXTAS.

An open-label study was carried out with citicoline (cytidine-5-diphospho-choline), a phospholipase A2 inhibitor after preliminary data suggested it was helpful in the *Drosophila* model of FXTAS [121]. Ten patients with FXTAS were given 1000 mg per day of citicoline for one year [426]. The primary outcome measure of improvement in the FXTAS rating scale, which is a quantitative measure of predominantly motor symptoms, was not met, although the patients were stable over the one-year study. Secondary-outcome analysis demonstrated some improvements on the Stroop assessment of executive function and in an anxiety measure.

Sulforaphane is a sulfur-containing phytoprotein found in the seeds and plants of cruciform vegetables, including broccoli, cauliflower, and Brussel sprouts, that impacts mitochondrial function through both Nrf2-dependent and independent mechanisms. Based on the positive effects of sulforaphane in a fibroblast model of FXTAS [329], an open-label trial (*n* = 15 subjects) was performed with oral sulforaphane (Avmacol). Eleven patients completed this trial, and no significant effects were seen on quantitative tremor or ataxia measures, which were the prespecified primary outcomes. However, some benefits were seen in secondary measures, including neuropsychological testing and in some molecular biomarkers. These benefits were described in more detail at the meeting and in a manuscript describing the effects [427].

### 9.2. Management of Neurologic Symptoms in FXTAS

Much of FXTAS management currently focuses on pharmacological and nonpharmacological strategies to address these symptoms, which are highly variable between individuals. As with many other ataxic disorders, there are few randomized placebo-controlled trials of interventions in FXTAS patients, and, as such, symptomatic treatment is prescribed based on the signs and concerns of the patient using medications approved to treat similar disorders [428].

There are three tremor patterns in FXTAS [232]. The originally described tremor is a symmetric action tremor of the hands, most similar to that seen in the essential tremor (ET). Propranolol and primidone have both been used in FXTAS and can be given as monotherapy or combined. Primidone needs to be used with caution, as it can worsen balance in patients with FXTAS, as seen in other disorders. Gabapentin may be useful as a third-line tremor agent, as it has been shown in small case series to potentially be helpful for ataxia. Topiramate is less useful given its cognitive side effects. Additional practical measures, such as the provision of weighted utensils or cups with a lid or straw, can also help with improving tremor management. Rest tremor can be seen in FXTAS, and medications used in Parkinson’s disease can be effective. Amantadine may help tremor and improve ataxia. Carbidopa/levodopa is effective in many FXTAS patients, especially when other parkinsonian signs are present. A third type of tremor, manifesting as a variable, higher amplitude, cerebellar (rubral-like) tremor can also be present and is more challenging to treat. Levetiracetam or zonisamide may be more effective in these patients.

A subset of FXTAS patients who have failed these agents have undergone unilateral or bilateral deep brain stimulation (DBS) in the ventral intermediate nucleus in the thalamus [429]. While some case reports suggest improvements with this measure, anecdotal reports of clinical decline in FXTAS patients after bilateral DBS reduced the use of this procedure in this population. More recently, unilateral DBS has been used and has shown less worsening of ataxia and cognition after surgery. MRI-guided focused ultrasound may also be used for unilateral tremor, but there are only case reports published, and the risk of side effects post procedure are still uncertain.

Treatment of ataxia is more challenging, and medications used for other ataxias may be considered [430]. FXTAS patients were included in a clinical trial that showed riluzole probably improves ataxia signs at 12 months. Medications that are used in other ataxias, such as 4-aminopyridine, or other gait disorders, such as amantadine, can be effective in isolated cases. Physical therapy is a mainstay of the management of imbalance in FXTAS. Many patients require use of an assistive device as their disease worsens, with a proportion of patients becoming wheelchair bound.

Executive dysfunction and cognitive decline are common symptoms in FXTAS. As such, FXTAS patients are often tried on agents used in other dementias [423]. Some patients respond to acetylcholinesterase inhibitors, such as donepezil, if they are experiencing memory loss as a significant feature. Memantine is also often tried based on the limited data suggesting physiological improvement in secondary measures in clinical trials, despite a lack of efficacy in memory-outcome measures [400,431]. Less commonly, FXTAS patients will develop psychosis as their cognition worsens, and quetiapine can be used.

In addition, FXTAS patients with neuropathy often do not complain of pain or even recognize the presence of numbness in their feet, although many do [34]. For those patients with symptoms, gabapentin is a good first-line treatment, as it can be also beneficial for tremor and balance. For both cognitive symptoms or neuropathy, an appropriate workup should be conducted in FXTAS patients to rule out other reversible causes of their symptoms, as treatment for those disorders may also be helpful for symptomatic improvement.

### 9.3. Lifestyle Changes in FXTAS

The Lancet Commission on Dementia identified 12 modifiable risk factors which explain 40% of the incidence of dementia [432]. This included education, hearing loss, traumatic brain injury, hypertension, excessive alcohol, obesity, smoking, depression, social isolation, physical inactivity, air pollution, and diabetes. How much these modifiable lifestyle factors would apply to slow the development and progression of FXTAS has not been specifically studied, but presumably could play a significant overlapping role. Exercise and diet seem to be the major factors in reducing mortality. When both aerobic and resistance exercise are optimally combined, systematic analyses have shown that the mortality rate from all causes can be reduced by around 30% [433]. Similarly, both aerobic and resistance exercises appear to improve cognitive abilities for older adults [434]. This appears to be partly due to the release of BDNF (brain-derived nerve growth factor) due to exercise, which stimulates neuronal growth and survival [435]. Resistance exercises and resistance training have been reported to evoke substantial functional brain changes, especially in the frontal lobe, accompanied by improvements in executive functions [436]. Furthermore, resistance training led to lower white-matter atrophy and smaller white-matter-lesion volumes. Controlled studies are needed for individuals at risk of or developing FXTAS to see if exercise can reduce these problems.

A healthy diet has also been shown to reduce the development of chronic diseases and mortality risk [437]. Recommendations from the USDA and HHS for a healthy diet include vegetables, fruits, grains, dairy, protein foods, and oils [438].

### 9.4. Future Directions to Advance Treatment in FXTAS

The most significant risk factor for cognitive decline and dementia is age. Rapamycin (Rp), also known as Sirolimus, is a drug that extends lifespan in various animals by inhibiting mTOR, a growth-regulating kinase. Rp, considered safe and available for years, has applications in organ transplants and medical coatings. It binds to an intermediary binding protein (FKBP12), inhibiting mTOR complex 1 and 2; intermittent dosing counters mTORC2 inhibition [439,440,441,442,443,444].

A 2006 Rp study on an FXTAS fly model that had a CGG90 attached to a green florescent protein (EGFP) reported that Rp worsened measures of degeneration in the eye, which were characterized by loss of pigmentation, neuron death (escharosis), retinal collapse, and ommatidial fusion [445]. A 2015 study confirmed this finding in flies [173]. However, in both studies, Rp was given continuously via daily food. Therefore, the adverse effects of chronic mTORC2 inhibition would not have been overcome [446]. Matt Kaeberlein has argued that is time for a controlled trial of Rp in humans using an intermittent dosing protocol to slow the rate of aging [447]. It seems it is time for an intermittent dosing Rp trial in the mouse model of FXTAS, and if successful, then in patients with FXTAS.

Potential future molecular therapies for CGG-repeat disorders and FXTAS have been discussed, which include ASOs, RNA interference (RNAi), small molecules, and gene editing using CRISPR approaches [138,145,448]. Some of these approaches are currently being clinically trialed in amyotrophic lateral sclerosis (ALS), myotonic dystrophy I, Huntington’s disease, and frontotemporal dementia, but for FXTAS, there are still several technical difficulties that need to be overcome. A recent study, however, did point the way to a potential small molecule [146].

Peng Jin and collaborators performed whole-genome sequencing (WGS) on male PM carriers to screen for candidate genetic modifiers. They found 18 genes that, when knocked down, were potential modifiers in a fly (*Drosophila*) model of FXTAS. The knockdown of one specific gene, Prosbeta5 (PSMB5), suppressed CGG-associated neurotoxicity in the fly model [449]. They also found that a polymorphic variant of PSMB5 with a human control allele frequency of 0.0625 was associated with the delayed onset of FXTAS in PM carriers. PSMB5 is a proteosome inhibitor, and several such proteosome inhibitors (bortezomib, carfilzomib, and ixazomib) are FDA-approved for the treatment of multiple myeloma. They discovered that one, ixazomib, reduced the expression of PSMB5 in both the fly model and human cells. This interesting finding offers a potential small-molecule therapy for FXTAS, but it awaits critical in vivo follow-up studies in mouse-FXTAS-model systems. Finally, sufficient human data on PSMB5’s effects in FXTAS are lacking.

According to recent research, there may be pharmacological interventions that can mitigate the progression of neurodegenerative disorders in their early stages. The U.S. FDA has granted accelerated approval for two medications (Lecanemab and Aducanuma) that could potentially slow the progression of AD in individuals with mild disease, although their efficacy in those with early AD is still under investigation [450]. Therefore, early diagnosis and intervention with such medications could potentially delay or mitigate the severity of FXTAS.

## 10. Screening for Fragile X and FXPAC

Individuals with a PM may find out their status through multiple potential avenues. The most common pathway to diagnosis has historically been through the diagnosis of a family member with FXS, resulting in cascade testing for at-risk family members. However, advances in genetic testing have driven costs down and increased awareness and guidance for screening at various stages. As a result, the landscape of screening has changed over the last decade. This section outlines the current status of screening at various time points, including carrier, prenatal, and newborn screening efforts.

### 10.1. Diagnosis via Cascade Testing

Current guidelines for testing focus on affected individuals, most commonly children with developmental delays and/or autism. Testing for FXS is recommended by the American Academy of Pediatrics in the United States for any individual who presents with ID, developmental delay, autism, family history, or other features common in FXS. Testing for the presence of the *FMR1* mutation is recommended for patients with cerebellar ataxia and intention tremors, especially if they are male, and for women who have ovarian insufficiency or elevated FSH levels.

Despite these guidelines and increased advocacy efforts to bring awareness to physicians about fragile X, an extensive diagnostic odyssey is still common. Boys, who are the most severely affected by FXS, are diagnosed around age 3 on average; girls with FXS, on the other hand, are on average diagnosed later, and individuals with a PM are rarely diagnosed in childhood. Fewer than 20% of children with symptoms received a diagnosis within a year of first concerns [451]. Around a quarter of families with a child with FXS will have a second child with FXS before the first is diagnosed [452].

Once a child with FXS has been diagnosed, genetic-counseling recommendations include testing of all immediate family members who are at risk for an expanded *FMR1* gene based on hereditary patterns [453]. Mothers of children with FXS are obligate carriers, but screening is necessary to determine the size of the mutation (full vs. PM) and for guidance regarding their own health risks.

In addition to the biological mother of the child with FXS, testing is recommended for siblings of the proband, as well as the maternal grandparents if available, and any siblings, nieces, and nephews of the mother. However, information about genetic risk to family members is often left to the affected individual to communicate, which may only occur in a small percentage of cases [454].

Finally, in addition to guidelines focused on testing of symptomatic children, there are also those regarding diagnostic testing for individuals exhibiting symptoms of FXPOI or FXTAS [455].

### 10.2. Newborn Screening

Newborn screening for FXS has been discussed for well over a decade, and although screening all newborns shortly after birth is supported by many developmental and behavioral pediatricians [456], FXS is not one of the standard conditions that states screen for based on the Recommended Uniform Screening Panel set forth by the Secretary of the Department of Health and Human Services. FXS does not meet current criteria for inclusion in newborn screening panels; namely, the requirement to have a proven treatment that needs to be implemented early in life [457]. Thus, newborn screening for fragile X is conducted after parental consent. Research studies have shown moderate-to-high acceptance rates by parents, ranging from about 62% to 94% [458,459]. Diagnosis shortly after birth has enabled work to understand early symptom onset, which helps to document the natural history of fragile X [70,460]. Although barriers continue to exist for full-scale newborn screening for fragile X [295,461], including the need to document early treatment benefit, there have been improvements in high-throughput screening techniques [462,463].

There have been several pilot studies testing the acceptability and feasibility of screening for FXS in the newborn period [458,464,465]. The most recent pilot, and one discussed at the conference, is the Early Check [143], an innovative, expanded newborn-screening program that offers voluntary expanded newborn screening to every mother in North Carolina. *FMR1* expansions were included as a part of the Early Check program from 2018 to 2021, and included screening for a full mutation along with an optional opportunity to receive PM results.

Raspa and colleagues explained at the conference that, over the course of the 38-month pilot, 18,923 infants were screened, with 6 infants identified with a full mutation. Of relevance to the PM field was the finding that around half of parents who consented to find out about FXS also consented to find out about the PM (*n* = 9550). Preliminary data presented at the conference showed that a total of 94 infants with a PM have been identified thus far. The range of CGG repeats was 54–115; two female infants with a PM were found to have a fully methylated allele. All of the families of infants with FXS, and the majority of those with a PM, opted to participate in developmental follow-up, which is offered through the child’s third birthday [143].

This is a significant achievement for the field, as it establishes a novel pathway to understanding the natural history of the PM clinical impact from birth. This is important because, due to the limitations of testing infants and toddlers for PMs, the recruitment of infants into research can be challenging. The published studies that do exist suggest that the effect in these early years may be tuned to nonverbal communication, sensory responses, fine motor development, and spatiotemporal processing [460,466,467]. There is no evidence in infancy for any impact to cognitive development, adaptive behavior, temperament, or overall communication skills [460,467]. As discussed at the conference, few infants and toddlers with a PM in the Early Check study are showing signs of overt developmental delays. However, of the 34 infants actively engaged in the follow-up study, 4 (12%) qualified for early intervention services due to delays in language and/or motor development. Further, on average, boys with a PM in the study are reported to have elevated concerns with oral processing (e.g., oral–sensory or oral–motor problems that interfere with feeding/eating), repetitive motor movements, ritualized and routinized behaviors, and/or restricted interests [143].

### 10.3. Carrier and Prenatal Screening

When asked about a preferred time, preconception carrier screening was favored by 76% of women over prenatal screening of the fetus, newborn screening, or when problems occur [468]. Screening prior to pregnancy allows for more informed decision making about reproductive options [469,470].

Because the presence of a PM can lead to developmental disability in future generations, carrier screening has important reproductive and mental health implications for the woman who has to be tested, and it is relevant for early detection, intervention, and family planning. A recent study investigating the feasibility of fragile X carrier screening for pregnant women and for the fetuses during prenatal diagnosis reported that, over a total of 7000 pregnant women, approximately 61% of them consented to receive carrier screening [471]. The study identified five PM carriers and three women with a full mutation, and suggested that implementation of carrier screening during prenatal diagnosis may provide information about early intervention for those who have the *FMR1* mutation, and information on the risk of having a child with FXS in the following pregnancies. Importantly, the identification of women carriers of a PM who are at risk for FXPOI could lead to more effective reproductive interventions for those who want to have a child. In addition, it has been reported that prenatal screening is beneficial when considering the burden of raising a child with FXS [471].

However, the debate continues over who should be offered prenatal carrier screening, currently available via chorionic villus sampling and amniocentesis, for a number of reasons, including concerns around counseling for this complex condition, for informed decision making, and the potential for the psychosocial aspects of screening [472,473].

### 10.4. Genetic Testing Pathways

An actively debated topic at the conference was regarding current genetic testing pathways in children. Three compelling case studies presented at the conference demonstrated gaps in the current system that may lead to the missed or incorrect diagnosis of an individual who has a PM [143]. This generated robust discussion about indications for testing for a PM within FXS-affected families, what symptoms to look out for in PM children or untested siblings who may have a PM, and of potential gaps in the applied diagnostic workflow. Regarding testing indication (i.e., the reason for testing a child for a PM), the current approach recommends testing in the instance of a clinical phenotype of developmental delay, ID, ASD, or multiple congenital abnormalities [474,475]. It was agreed at the conference that this pathway needs some refinement, particularly around the phenotypic indicators for testing, and further research into appropriate models that do not exclude PM children is warranted. One emerging concept is that, because subtle signs are easily missed in children with a PM, apparently asymptomatic or unaffected children may be considered for testing and follow-up neuropsychological assessment on a case-by-case basis.

Potential limitations of the diagnostic workflow typically employed in the clinical pediatric setting were also discussed (i.e., microarray and CGG sizing that may be followed by Southern blot) [476]. One issue discussed was that, in the case of cascade testing, microarray may be skipped, and the workflow may be directed to *FMR1* testing only. If a PM is then detected, it is possible to overinterpret the common PM as causative when it is potentially incidental, and there may be another finding on the microarray or another platform that has been missed. This is potentially clinically important, given that Lozano et al. [44] found that further genetic testing with a microarray or WES in people who have a PM with ID or ASD demonstrated that 20% had a second genetic hit that likely exacerbated the PM phenotype, resulting in more severe involvement.

It is also notable that PM alleles may be unstable and differ across tissue types [86,90,95,477,478], and the repeat size mosaicism of PM/full-mutation alleles is more likely to occur when the predominant PM CGG repeat length is at the upper end of the PM range [479]. This implies that the CGG expansion can be present at a larger size (e.g., a full mutation) in unmeasured tissues (including the brain). Indeed, given new data on instability (reviewed in the molecular section of this publication), a PM may change in size over time. Instability in the PM allele in females has also been associated with the clinical phenotype of ADHD, and this was reported at the conference and is now published [86,91]. Thus, some flexibility in understanding the blurring between the PM and full mutation, and the possibility for additional findings not routinely detectable with standard of care testing workflows, is recommended.

## 11. Shining a Light on the *FMR1* PM: What We Know, What We Think We Know, and What We Need to Know

The conference demonstrated a critical step forward in the inclusion of a lived-experience perspective of those with the PM. This was in the form of inclusion of the consumer voice in group discussions, as well as two presentations from representatives of the FXS and PM community. Following presentations, discussion, and at times vigorous debate, a range of themes emerged at the conference. These included the importance of population screening and the information shared with individuals newly identified with the PM; development and use of terminology in this emerging field of study and the need for agreed, consistent use of terminology for both individuals with the PM, clinicians, and researchers; the concept of ‘at increased risk of’ when considering how to talk about the range of issues associated with the PM; recognizing the PM population currently studied is skewed towards families impacted by FXS; the importance of the lived-experience voice. The quantity and quality of research shared was impressive and highlighted the evolving understanding of what we know, what we think we know, and what we need to know about the PM.

Data from studies presented at the conference extend the current literature that has investigated health impacts linked to the PM. This includes FXPAC, FXPOI, FXTAS, and FXAND. These conditions range from subtle effects, that in some cases are difficult to measure, through to clinically diagnosed conditions. Issues identified in studies as having higher prevalence than the general population included anxiety, depression, executive function difficulties, autoimmune conditions, hypothyroidism, migraines, chronic pain, and sleep apnea. There is a recognized need for early diagnosis and management of these conditions. However, this is not necessarily occurring due to lack of awareness amongst healthcare providers about the broader impacts of the PM. Further research will be instrumental in elucidating and defining these effects on health, and developing strategies to improve support for people with the PM.

Other areas of significance discussed during the conference included:Fertility-related issues—the need for increased knowledge and better pathways for fertility-related issues associated with the PM gene, particularly for younger women;The CGG repeat number is recognized as only part of the evolving picture—research indicating AR, *FMR1* mRNA, FMRP levels, AGG interruptions, and allelic instability as also important factors to consider;Lifestyle measures—multiple presenters mentioned the importance of healthy lifestyle as a protective measure against risk factors associated with the PM, including an emphasis on limiting alcohol, not smoking, the importance of exercise and a good diet, and avoiding excess environmental toxins and high stress;It was noted that many PMs have high levels of functioning and achievement;Many PMs also face the challenges of children with developmental issues and FXS.

There was much discussion about how to talk about the influence of the PM that may occur outside FXPOI and FXTAS. The concept of ‘at increased risk of’ was widely discussed as a way to share what is currently known about the health impacts associated with the PM. Talking about these conditions/effects as risk factors takes into account that there are still unknown elements, including the differences between males and females, the impact of CGG sizes, AR, FMRP levels, AGG interruptions, and environmental factors. Discussing the possible effects of the PM as elevated risk factors, compared to the general population, as opposed to labeling conditions, was an approach which received wide agreement.

Results from large-scale PM-reproductive-carrier screening in Australia has provided important information about the distribution of *FMR1* alleles in the general population. Approximately 75% of females with a PM in this cohort had 55–69 CGG repeats [480]. It is likely our understanding of the PM will evolve, and more research is needed as we widen the scope of research to increasingly capture those in the 55–69 CGG range. It was therefore recognized that our knowledge about the significance of the *FMR1* alleles is possibly biased, coming mostly from families impacted by FXS. Clinicians raised the issue of what information is shared with those newly diagnosed with the PM, acknowledging most research to date has focused on individuals with families impacted by FXS and, by implication, predominately those above the 55–69 CGG repeat range.

## 12. NZ Fragile X Community Response to PM Research (Fragile X New Zealand)

Data were captured from 38 people from the NZ PM community via a Survey Monkey questionnaire. The results showed the community is: (a) interested in PM research; (b) experiencing a high personal value from the research; (c) able to easily access and digest the research; (d) making positive changes in lifestyle choice and behavior based on recommendations from specialists in the field (Figure 4a). These data emphasized that the knowledge gained through research is valuable to persons with a lived experience and is enabling informed choices [143].

In recent years, there has been debate about proposed terminology encompassing the broader health impacts of the PM, with the terms FXAND and FXPAC both used. As shown in Figure 4b, the preferred terminology in the survey respondents was FXPAC. As demonstrated by the quotes, there are many positive experiences from this community, and recognizing the possible impact of the PM is valued (Figure 4c), while others outlined concerns about negative terminology (Figure 4d, quote 1). The results suggest a wide variety of personal experiences and opinions amongst the community and highlight the importance of ongoing research in this space that includes the voices of those with lived experience [143].

## 13. What Is in a Name? (National Fragile X Foundation (NFXF) and Fragile X Association of Australia (FXAA))

### 13.1. Terminology

The focus of this talk was on terminology, and to understand this more thoroughly, the team conducted a survey about the terminology related to the PM. Data were captured from 296 people from the NFXF and FXAA communities via a Survey Monkey questionnaire, 255 of whom reported having the PM.

Part 1 of the survey probed what terminology people with lived experience endorsed when talking about their own PM status. A “click all that applies” option was provided. The response with the highest endorsement was Fragile X Carrier (*n* = 148), followed by Fragile X PM Carrier (*n* = 110), and Fragile X PM (*n* = 85).

Part 2 of the survey delved into views as measured by a Likert scale ranging from very comfortable to very uncomfortable, about specific words, focusing on “Fragile X premutation carrier”, “Fragile X carrier”, “condition”, “conditions and disorders”, and “disorder”. As shown in Figure 5, not one of these options has strong support (~less than half of respondents were very or somewhat comfortable with these options). In the last section (part 3), an open text box was enabled to collect comments. These qualitative data demonstrate that the concerns of the community are, indeed, far more complex than initially assumed, extending to the words “premutation” and “carrier”, and this generated robust debate between the audience and speakers, which is still ongoing [143].

### 13.2. The Importance of Appropriate Terminology

Both surveys acknowledged the impact of language and terminology in the community. Both surveys highlighted challenges with terms such as ‘carrier’, which could imply that a person experiences no impact from the PM. Also identified were issues with terms such as ‘mutation’ and ‘PM’, and the desire for the use of neutral, nonstigmatizing language. The group reflected on learnings from other expansion condition fields, such as Huntington’s and myotonic dystrophy, which have also undergone phenotype and/or terminology updates. For example, the term ‘gene change’ is now used rather than mutation. In addition, the term most often used is “variant”. The variant may be neutral or increase the function of the gene (gain-of-function mutation) or decrease or inhibit its function (loss-of-function mutation). It was also noted in the autism field that the term disorder is no longer widely used. Both clinicians and researchers shared the need for consistent use of agreed language and terminology. It was recognized there was a focus by U.S. clinicians on the use of terminology that helps families and patients gain access to services they need in the U.S. Clinicians raised the challenges for practicing clinicians worldwide when terminology evolves and changes.

Fragile X International (FraXI) released a press statement during the International Premutation Conference sharing the position of 17 family led fragile X country organizations regarding the use of terminology. FraXI and many family led organizations have adopted the term Fragile X-premutation-associated conditions (FXPAC), a term which lists everything which may or may not affect a PM carrier, and aims to use neutral language which is nondiscriminatory. Dr. Randi Hagerman, in her summary of the discussion around premutation terminology, shared her view that the term FXPAC presented a sensible umbrella term to encompass the range of involvement from the PM, with FXAND sitting under the FXPAC umbrella.

## 14. Summary

We know more than we ever have about the *FMR1* PM. However, there remains more to learn and understand. This is particularly important as screening becomes increasingly available and those diagnosed with the PM seek to understand what this means, and the implications. Research suggests that some people with the PM may experience health impacts outside the currently defined FXPOI and FXTAS. Using appropriate, consistent, and nonstigmatizing terminology was recognized as having important implications for the planned knowledge translation of these new findings. This includes the potential success of developed guidelines for testing for both adults and in childhood that aims to provide early detection to inform optimal management and outcomes. Emerging from the International Premutation Conference, where many experts in the field met (Figure 6) and discussed the related issues, was an overall agreement around the value of the concept of ‘at increased risk’ compared to the general population when referring to the range of conditions currently associated with the PM.

For those researchers who are interested in enrolling people with the PM into their research, there is an International Fragile X PM Registry at: https://fragilex.org/our-research/projects/PM-registry/ (access date: 11 September 2023). It is also important to encourage your patients with the PM to join this registry.

## Figures and Tables

**Figure 1 cells-12-02330-f001:**
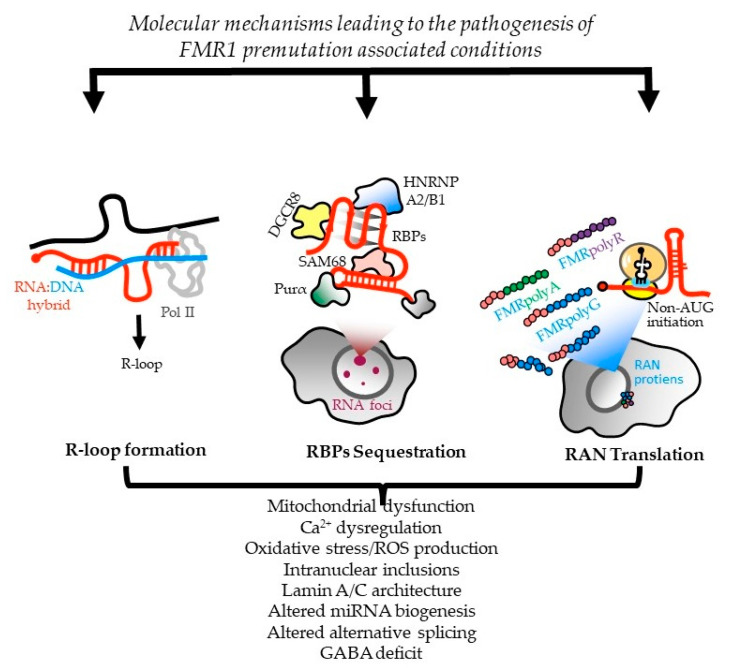
Molecular mechanisms leading to *FMR1*-PM-associated conditions. Three nonexclusive models are proposed for how CGG repeats contribute to the pathogenesis of PM conditions, including FXTAS. First, the cotranscriptional R-loop formation, which compromises genomic stability and triggers a DNA-damage response that can activate inflammatory cascades [116,117]. Second, CGG-repeat RNAs can elicit a gain-of-function toxicity through RNA gelation into nuclear foci and sequestration of various rCGG-repeat-binding proteins, leading to their functional depletion [25,26]. Third, repeat-associated non-AUG (RAN)-initiated translation generates potentially toxic proteins that accumulate within intranuclear neuronal inclusions in FXTAS patients. The relative contribution from each mechanism to downstream sequelae, such as mitochondrial dysfunction and neuronal death, and their potential synergies in disease pathogenesis, are areas of ongoing research in the field. Adapted from [118].

**Figure 2 cells-12-02330-f002:**
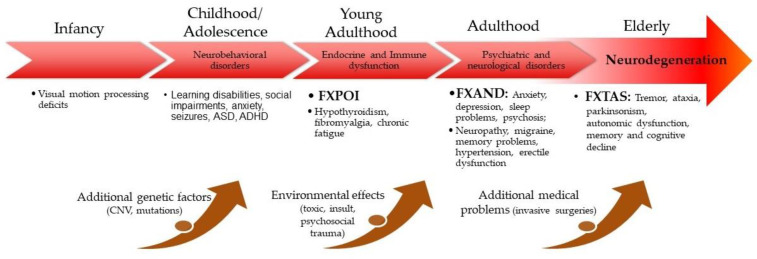
FXPAC involvement across the lifespan.

**Figure 3 cells-12-02330-f003:**
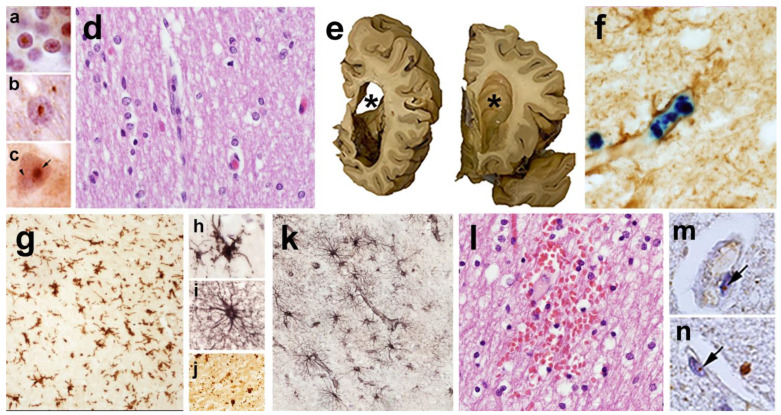
(**a**–**c**) H&E and ubiquitin staining (brown). Inclusions in astrocytes (**a**), neurons (**b**), and Purkinje cells (**c**); (**d**) H&E. White-matter disease in cerebellum; (**e**) Cortical atrophy and venrticulomegalia; (**f**) Perl’s staining. Iron deposition in capillaries; (**g**,**h**) Iba1 staining. Activated microglia; (**i**,**k**) GFAP staining. Activated astrocytes; (**j**) Iba1 staining. Senescent microglia; (**l**) H&E. Microbleeding; (**m**,**n**) H&E and ubiquitin staining (brown). Inclusions in endothelial cells. Arrows and asterisks are indicating the pathology of interest. Arrowhead in (**c**) points to nucleolus.

**Figure 4 cells-12-02330-f004:**
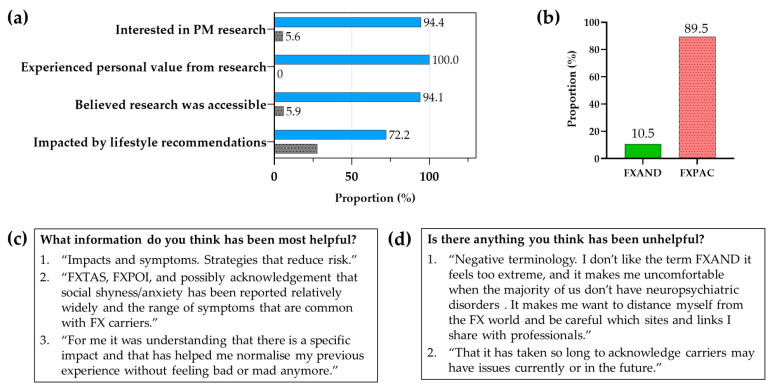
Lived experience perspective. Results from an anonymous Survey Monkey questionnaire distributed via email to the member based of Fragile X New Zealand (*n* = 38). (**a**) Binary data (blue = yes; gray = no); (**b**) Preferred terminology by survey respondents; (**c**,**d**). Example quotes in response to query about (**c**) what is helpful in research and (**d**) what is unhelpful in research.

**Figure 5 cells-12-02330-f005:**
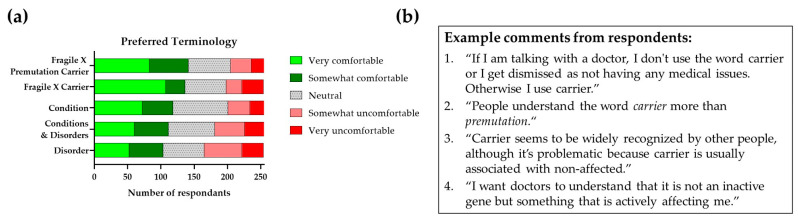
Lived experience perspectives from member based of NFXF and FXAA. (**a**) Results from an anonymous Survey Monkey questionnaire distributed via email to the member based of NFXF and FXAA (*n* = 255); (**b**) Example quotes.

**Figure 6 cells-12-02330-f006:**
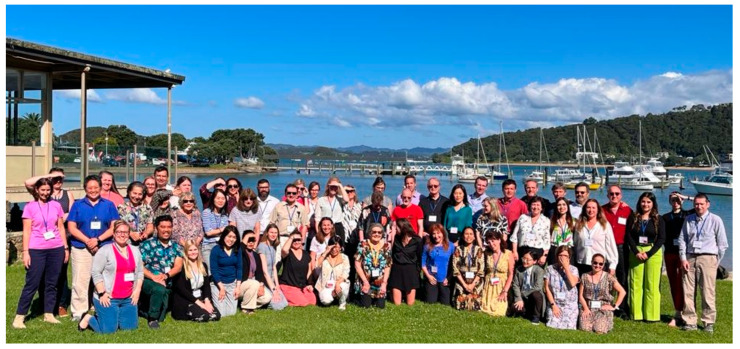
Attendees at the 5th International Conference on *FMR1* Premutation.

## Data Availability

Not applicable.

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
