# Peer review of "Insight and Recommendations for Fragile X-Premutation-Associated Conditions from the Fifth International Conference on FMR1 Premutation"

_cells, 2023, doi:10.3390/cells12182330_

Round 1

Reviewer 1 Report

This huge article is well written, I expect it will be widely studied and followed by communities interested in its pathology.

Author Response

We thank this reviewer for the encouraging and positive comments.

Author Response

We thank you the reviewers for their comments and suggestions which have greatly improved the manuscript

Please find below are our responses:

REVIEWER 2

  1. General

This manuscript is a conference report written by 46 authors. Instead of a special issue with separate abstracts and/or articles the journal’s Editor-in-chief has decided to allow this kind of a large article, a report, which is not a systematic review of what is known at the moment.

Novel data are presented at least on these lines:

282-286, 293-296, 306-309, 337-344, 458-463, 488-491, 543-546, 718-722, 770-779, 797-803, 966-973, 1105-1110, 1114, 1166-.,1218,11535,1365, 1458-, 1537-, 1583-, 1600-, 1706, 1726-, 1752, 1897-, 2065-, 2099-, 2318- and 2490-. Details of new data are not shown, materials and methods are not presented in most cases, and the discussion of advantages and weaknesses of results is lacking. The authors may take it for granted that the results they report have all been reliably obtained. In this review I pinpoint some examples showing that the results – either preliminary or final – can be controversial and the interpretation of the findings too straightforward and simplified. There are extrapolations ranging from Drosophila or cell cultures directly to the clinics without enough caution and doubt.

In general, there are much detailed data on genes and metabolism, somewhat demanding to a normal reader. Sentences are often very long. I recommend the authors to take a challenge of writing shorter and less complicated sentences. I also urge to make use of tables when presenting existing studies of a certain heading to facilitate comprehension and comparisons.

The need for future research is expressed at the end of every chapter, with unnecessary repetition.

Detailed criticism and questions

Thank you for the comments and suggestions. We have modified the manuscript accordingly where applicable and left some sections. This article is a very comprehensive review on the various aspects of FMR1 associated conditions and meant  for scientists with basic scientific knowledge and not for the general public.

  1. Introduction
  2. Could you insert a box explaining the terms FMR1, FXS, FXPOI, FXTAS and FXAND, and possibly a drawing?

We do not believe that an additional table/box to explain the terms is needed but we have clearly spelled the names out.

  1. Lines 171-174 are complicated in the introduction, “including …such that…”such as. Shorten the sentence by ”…RNA toxicity, clogging of polysomes, RAN translation and mitochondrial dysfunction, see chapter 2.”

We have made the suggested changes.

  1. Lines 264-249 can be omitted and replaced by some words in acknowledgements.

We have made the suggested change.

  1. Molecular pathology
  2. Lines 277-286 : The role of bidirectional transcription and alternative splicing in PM is highly interesting.

We thank the reviewer for the comment.

  1. Lines 294-295: lacking details.

We have added a few references demonstrating the potential role that ASFMR1 play in the pathogenesis of FXTAS and also of DNA repair genes.

  1. Lines 306-314: Allelic instability …may relate to diverse phenotypes” , is there any evidence?

A recent study has been recently published about the clinical implications of somatic

allele expansion in premutation carriers and we have added it to the manuscript.

  1. Lines 338-341: Is there enough evidence to advise methylation status analysis for all females with either FXS or PM alleles?

The paper we submitted includes several references about studies that support the importance of taking in consideration methylation analysis in females in both females with a premutation or full mutation alleles.

  1. Fig. 1 good. More explanations in legend, please.

We have made the suggested changes.

  1. Lines 451-453: Propagation via exosomes? Explain shortly what this means.

We have made the suggested changes.

  1. Lines 454-461: Results are all from mouse experiments and partly unpublished thus impossible to evaluate-.

We have made the suggested changes.

  1. RAN translation is a discovery, the biology of which is complicated and not well known yet. Could you shorten a little e.g. omit lines 478-481?

Thank you for your suggestion. These lines are omitted in current version of article.

  1. Lines 482-509. Could you summarize the use of ASOs and ubiquitin -proteasome targeted therapeutic possibilities, and make this shorter. Drosophila experiments, cell-based-models and mouse experiments are still far from the clinic.

Thank you for your suggestion. Based on your comments, the sentence has been modified.

  1. In chapter 4.2 FXTAS Spectrum there are very long sentences: that one starting on line 1044 with “subclinical motor impairments” twice in the same sentence.

Sentence has been edited for more clarity.

In addition, in the same paragraph, the next sentence was: “The predominance of intention tremor in the absence of gait ataxia or typical changes in cerebellar peduncles in these carriers led to speculation regarding the existence of modifying factors that might be accountable for neuroprotection in specific brain locations, such as cerebellum”.

We are not sure what the reviewer is asking here.

  1. line 1073. Leave out “(including CGG repeat and FMR1 mRNA )” – not necessary for comprehension of the sentence with references.

The sentence was removed.

  1. Line 1088-1089: …identified higher CGG repeat number as… Word “number” missing.

The word “number” has been added.

  1. Lines 1128- Kluseks studies are suggesting “a neuroprotective role” of higher education. It may as well be that there is biologically no causality between those two, but instead the premutation causes a change in the brain function that mediates psychological changes due to increased risks.

Changes were made.

  1. I think a table would be easier to read and to learn to summarize previous work in this area. I recommend adding lists of existing results in table form in the manuscript.

We do not think that a table would be helpful and easier to read but we have clarified through the text.

  1. The importance of healthy life style is self-evident. Likewise the need for further studies, a wish that is repeated in almost every chapter e.g. lines 1280 – on substance abuse.

We have modified the paragraph accordingly and added a sentence about the importance of future study.

  1. Autism
  2. Divide the long sentence starting on line 1300 into two.

The long sentence has been divided as suggested.

  1. Line 1336: It is not acceptable to put the Abstract book of the conference in question as a reference (134).

This paper serves as a comprehensive compilation of the data presented and deliberated at the proceedings of the 5th International Conference on FMR1 Premutation. Its primary objective is to provide a concise summary of the entirety of our collective knowledge pertaining to the premutation of the FMR1 gene.

It is worth noting that while some of the data presented at the conference have been subsequently published, other findings remain in the process of submission to scientific journals or are currently under review by experts in the field. This paper aims to bridge the gap between the current state of knowledge and the ongoing research developments in the realm of FMR1 premutation.

  1. Other symptoms
  2. Metabolic changes are discussed starting from medical basics. Still in this chapter 4.5.2 there is just one reference. Chapter 4.5.2. is not convincing and too long.

We have clarified the potential metabolic risks while addressing the reviewer's feedback about chapter length and focus.

  1. Chronic pain and fibromyalgia, chapter 4.5.4. results are based on case reports. This makes the whole chapter speculative, not convincing.

The description of the cases reported in the past were removed and only referenced.

  1. Lines 1480-1499 are not on solid basis: What are the standardized definitions of “central sensitivity syndrome”? Lacking of concordance between patient and physician opinions makes the whole discussion very speculative. I don’t support publication of that part of the text in the present form. Case-control studies do not support the association of the premutation and these symptoms.

We have made the following edits:

A reference is included for the definition of central sensitivity syndromes used in the cited study (Yunus MB. Central sensitivity syndromes: a new paradigm and group nosology for fibromyalgia and overlapping conditions, and the related issue of disease versus illness. Semin Arthritis Rheum 2008: 37: 339–352).

Although case-control study designs may not be ideal to look at genotype-phenotype relationships, the cited study did involve specialty providers (neurologist, endocrinologist, and neuropsychologist) who were blinded to gene status doing the examinations and lab testing of the premutation carriers and controls. This provides some information to the reader regarding the specific pre-determined questions/aims of the study and will be of interest to a reader of this paper, even with caveats of this study design.

Additional text was added that the study authors noted the discrepancies and commented on the possible interpretation in the published paper, and these were not the speculations of the authors of this review paper.

  1. FXTAS clinical and protective mechanisms
  2. Line 1515: What do you mean here by the expression “in the 50s and 60s”?

We edited the sentence and clarified “…Having CGG repeats in the range from 50 to 60 repeats…”

  1. Line 1536: To be able to prevent FXTAS by lifestyle choices doesn’t have a sound scientific basis. The idea is highly speculative.

Thank you for your comment. We agree with your comment. Accordingly, we have changed this statement. Please, find previous and current statement.

  1. Line 1563 onwards: why does high educational level have an association with reduced FXTAS symptoms? I would not use causality-related terms here. This is an association. Or do you know the real brain physiological mechanisms? Higher education cannot be “neuroprotective” in a scientific sense, it can be an “environmental” factor, maybe epigenetic, associated with the described symptomatology but the mechanisms are not known.

We have made the suggested changes.

  1. Lines 1609-1629 ok.

Thank you.

  1. Psychotherapy treatment
  2. lines 1668-1687- The chapter starts with basics as in a medical textbook. References 350-358 don’t concern the FMR1 premutation symptomatology at all. This part can be shortened.

Thank you for your suggestion. We agree with your comment that references 350-368 don’t concern the FMR1 premutation symptomatology at all. This, we shortened this paragraph.

  1. Lines 1730-1751 there is much space given to the study of Shelby et al. without reference, so this is new data, suggesting omega-6 fatty acid and arachidonic acid metabolism is “altered” in FXPOI, but the only reference is to mouse experiments. Again “further studies are necessary…” which is evident. What would a biomarker -if discovered - give in this case?

We have edited the paragraph as suggested.

  1. Neuroimaging in FXTAS
  2. Lines 1814-1828. Were these studies performed blinded? Were there controls?

We have made changes and added requested data.

  1. Brain function
  2. Line 1884: The conclusion of “strong possibility” of “drug discovery” is an exaggeration not on a scientifically solid ground. Change wording.

Changes were made.

  1. Line 1914-1920: Too long sentence, divide in two or shorten.

The long sentence was divided as suggested.

  1. Lines 1921-1936 ok.

Thank you.

  1. Neuropathology
  2. Figure 3
  3. This is not a good picture, with PAD slides with different staining. The autopsied brain should be a completely different figure, and only if such big changes really occur in FXTAS brains. I think brain imaging by MRI might show white matter changes better.

Figure 3 has been modified.

  1. The staining in various histological samples are different. They are not described. The comparison of the slides is therefore impossible. The photographic quality - in the internet at least- is not satisfying.

We described the staining and images have been changed with higher resolution.

  1. Photographs 3d, 3f, 3g, and 3k do not show clearly what is ment to be seen. There is no arrow in 3c. There are no asterisks in the PAD slides. The four arrows in3d are pinpointing what?

Requested changes and corrections were made to Figure 3

  1. Explain “gemistocytic cell bodies” in the text on Line 2019

An explanation has been added to the manuscript

  1. Treatments
  2. line 2065-2079. These new data on memantine have much weight, could be shortened

The paragraph has been shortened in the current version.

  1. Line 2083: post-partum (spelling)

Post-partum was corrected as postpartum depression.

  1. Line 2096: Omit the web address. This is a commercial otherwise.

Web address was omitted.

  1. Lines 2099-2116: Data on citicholine and suphoraphane are preliminary, presented at the conference, not showing wanted effects and not convincing. It is unacceptable to have the Abstract book of the conference as a reference (134).

The data regarding an open label treatment study of citicholine  in FXTAS is described and cited in the text already and this has been published by Hall et al 2020 PLOS One and this is cited in the reference list already, reference 417. The information on sulforaphane open label study was presented at the conference and the paper has been submitted to Cells and is currently under review. We have now cited this paper in the reference 134 instead of the abstract book. We have included a mention of this study since it was presented at the conference and this paper is a summary of what was presented at the conference (Santos et al, 2023-under review).

  1. Lines 2125-2138: There are recommendations for medical treatment without any references. Are there randomized controlled trials on these treatments? How are these guidelines created or are they based on the authors’ (whose) own experience?

These medical recommendations pertain to treatments of tremor that represent standard of care for tremor cases in general. No controlled trials of these meds have been carried out in FXTAS however, the movement disorder neurologists including Drs Hall and Todd have considerable experience with FXTAS and these medications have been helpful in their FXTAS patients also.

  1. Line 2188-2197: Healthy diet is self-evident/textbook data. A reference is enough.

In current version, this part is shortened and presented in one sentence.

  1. Lines 2199-2212: The rapamycin chapter is introduced very extensively. Shorten.

The rapamycin chapter is shortened in current version.

  1. Line 2214: What exactly do you mean by “worsened a measure of neurodegeneration in the eye”? i. It is not scientifically sound nor ethical to announce the self-usage of rapamycin to prevent aging. Omit lines 2219-2231. The last sentence is acceptable.

We have edited the sentence for more clarity. Lines 2219-2231 have been omitted in the current version of the article.

  1. Lines 2240: Prosbeta5 is a myeloma drug, a proteosome inhibitor. Drosophila and mouse data exist in FXTAS models, but human data is lacking.

The sentence has been modified and “human data is lacking”has been added as suggested

Line 2244 Omit the words: “decreased quantitative-expression”, they make the sentence complicated

The words “decreased quantitative-expression” have been omitted.

  1. Screening
  2. Line 2338-2341: No reference given.

The reference has been added.

  1. Line 2358-2260: No reference given.

The reference has been added.

  1. Line 2375: “…may support cascade testing”. Testing of asymptomatic / unaffected children is in clinical genetics considered unethical as a general rule. Leave out this suggestion and the words “may benefit”.

Changes were made accordingly.

  1. Summary of the conference
  2. Lines 2318- This is kind of a new introduction. The “consumer voice” is one aspect that weakens the scientific, value of this report see below. On lines 2427-2437: The word “impact” is there at least three times. Replace one or two by “effect” or “influence” or some other synonymous word.

Mostly, the word “impact” has been replaced with synonymous.

  1. Lines 2435-2437: All children that MAY have learning problems need to get special education and support at school. It is not ethical to test children. Omit this sentence.

Sentence was omitted.

  1. Fig. 4. 4a) The legend does not say what the blue and black bars stand for.

4b) Add legend: preferred terminology by the survey respondents.

The Figures 4 and 5 legends have been updated for more clarity.

  1. Lines 2488-2534 Terminology, being a FRM1 carrier, either FRM1 premutation carrier or FMR1 full mutation carrier (=FXS female) are not identical or interchangeable as suggested by New Zealand survey audience.

Changes were made and new Figures 4 and 5 provided.

  1. Line 2521: “Gene change is now used instead of mutation” is not quite right, the term most often used are “variant”. The variant may be neutral or increase the function of the gene (gain-of-function mutation) or decrease or inhibit its function (loss-of-function mutation).

We have made the change. The term most often used are “variant”. The variant may be neutral or increase the function of the gene (gain-of-function mutation) or decrease or inhibit its function (loss-of-function mutation)”.

Reviewer 3 Report

This is a review article summarizing health issues associated with the fragile X premutation condition as well as recommendations discussed at the International Premutation Conference in New Zealand in 2023. The review is detailed and thorough with 79 pages and 469 citations. It is more a book, i.e., encyclopedia of fragile X-related disorders, than a review paper. It covers all known problems associated with the premutation including a detailed history of the findings leading to discovery of FXTAS and FXPOI, the molecular basis of FXPAC, clinical aspects in children with PM, FXPAC relationship with genetic markers, clinical and protective mechanisms of FXTAS, reproductive and health implications, FXTAS neuroimaging findings, FXTAS neuropathology, FXTAS treatment, screening and terminology. A thorough reading by someone in the field requires about 5 hours. Each section, which includes subsections, could be an independent review article.

Minor suggestions:

The manuscript would benefit from an abbreviation table.

Line 200-204: Lein lab work could be mentioned – PCB toxins and FXTAS mouse model

Line 489-490: “suggested” used twice in same sentence

Line 543: “data were” instead of “data was”

Line 568-578: Maybe a comment should be added about different activation ratios between tissue types as a potential confounding factor? These data are mentioned at line 2392.

Line 579 is confusing – “there are two types of pluripotent stem cell lines”. The authors seem to be combining embryonic and adult (induced pluripotent) stem cells into one category. I think the discussion of the different cell types should be clearly differentiated here as well as with the discussed results and potential use as “powerful tools”. The data discussed from lines 587 onwards all covers IPSC.

Line 606: not clear what type of stem cell is being referred to and a stretch to emphasize as a powerful tool to identify molecular events in FXTAS, which occurs decades later after significant environmental exposure and aging.

Line 1077: could define dyscalculia with first use.

2037: Parkinsonian should not be capitalized as it’s an adjective (not proper noun) and for consistency with other instances in manuscript

Line 2314: suggest terminology “mothers” instead of “birthing parent”

There is a shortage of discussion (probably due to lack of data) on the mechanism of increased intelligence with the premutation. Is there any research on the positive effects of the premutation as an adaptive advantage? Higher IQ? Decreased cancer? If so, should be discussed.

Author Response

We thank you the reviewers for their comments and suggestions which have greatly improved the manuscript

Please find below are our responses:

REVIEWER 3

This is a review article summarizing health issues associated with the fragile X premutation condition as well as recommendations discussed at the International Premutation Conference in New Zealand in 2023. The review is detailed and thorough with 79 pages and 469 citations. It is more a book, i.e., encyclopedia of fragile X-related disorders, than a review paper. It covers all known problems associated with the premutation including a detailed history of the findings leading to discovery of FXTAS and FXPOI, the molecular basis of FXPAC, clinical aspects in children with PM, FXPAC relationship with genetic markers, clinical and protective mechanisms of FXTAS, reproductive and health implications, FXTAS neuroimaging findings, FXTAS neuropathology, FXTAS treatment, screening and terminology. A thorough reading by someone in the field requires about 5 hours. Each section, which includes subsections, could be an independent review article.

Minor suggestions:

The manuscript would benefit from an abbreviation table.

Thank you for your suggestion. Please find the abbreviation table at the end of the manuscript.

Line 200-204: Lein lab work could be mentioned – PCB toxins and FXTAS mouse model

References were added.

Line 489-490: “suggested” used twice in same sentence

The sentence was edited.

Line 543: “data were” instead of “data was”.

Correction was made.

Line 568-578: Maybe a comment should be added about different activation ratios between tissue types as a potential confounding factor? These data are mentioned at line 2392.

It is not clear to the authors were in the manuscript this should be clarified.

Line 579 is confusing – “there are two types of pluripotent stem cell lines”. The authors seem to be combining embryonic and adult (induced pluripotent) stem cells into one category. I think the discussion of the different cell types should be clearly differentiated here as well as with the discussed results and potential use as “powerful tools”. The data discussed from lines 587 onwards all covers IPSC.

We are combining ESCs and iPSCs into one single category based on their potency (both pluripotent) and not on their tissue of origin, since we are talking about human pluripotent stem cell-based models of FXTAS. However, if the literature about the use of iPSCs to model FXTAS is consistent, very little is available for models based on hESCs (Gerhardt et al., 2014). So, to be more precise we added reference Gerhardt et al., 2014 and accordingly made changes.  

Line 606: not clear what type of stem cell is being referred to and a stretch to emphasize as a powerful tool to identify molecular events in FXTAS, which occurs decades later after significant environmental exposure and aging.

We agree with the reviewer that we are talking about a neurodegenerative condition that shows phenotypic manifestations in the elderly life.  However, there are several reports that we cited in which iPSC-based models have been instrumental in dissecting some FXTAS-specific disease mechanisms. What we are pointing out here is that increasing the complexity of our human-specific model starting from PSCs (and, in particular iPSCs) could allow for studying the interactions among different cell types in the context of a developing brain in which the clinical manifestations are not already visible. This gives us the advantage of dissecting the pathogenic mechanism at very early stages. We did not specify iPSCs or ESCs on purpose since, in principle, both cell types could be used to model FXTAS. However, to be more specific, since we cited only iPSC-based models, we converted PSC with iPSC.

Line 1077: could define dyscalculia with first use.

We defined dyscalculia

2037: Parkinsonian should not be capitalized as it’s an adjective (not proper noun) and for consistency with other instances in manuscript,

Changes were made.

Line 2314: suggest terminology “mothers” instead of “birthing parent”

Correction was made.

 There is a shortage of discussion (probably due to lack of data) on the mechanism of increased intelligence with the premutation. Is there any research on the positive effects of the premutation as an adaptive advantage? Higher IQ? Decreased cancer? If so, should be discussed.

There is no data about the mechanism of increased intelligence with the premutation nor about the adaptive advantage in carriers. There is no data about decreased cancer in the premutation and in fact the opposite may be true because an increase in FMRP is associated with cancer from Claudia Bagni’s work.

Round 2
